# The design of unit cells by combining the self-reproduction systems and metabolic cushioning loads

Kristo Abner[1,2,3], Peter Šverns[1,2], Janar Arold[1,2], Taivo Lints [1,2], Neeme-Andreas Eller[1,2], Indrek Morell[1,2], Andrus Seiman[1,2], Kaarel Adamberg[1,2] & Raivo Vilu [1,2,3] ✉

Recently, we published a comprehensive theoretical analysis of the self-reproduction processes in proto-cells (doubling of their components) composed of different combinations of cellular subsystems. In this paper, we extend the detailed analysis of structural and functional peculiarities of self-reproduction processes to unit cells of the Cooper-Helmstetter-Donachie cell cycle theory. We show that: 1. Our modelling framework allows to calculate physiological parameters (numbers of cell components, flux patterns, cellular composition, etc.) of unit cells, including also unit cell mass that determines the DNA replication initiation conditions. 2. Unit cells might have additional cell (cushioning) components that are responsible not only for carrying out various special functions, but also for regulating cell size and stabilizing the growth of cells. 3. The optimal productivity of the synthesis of cushioning components (useful cellular load) is observed at doubling time approximately two times longer than the minimal doubling time of the unit cells.

Self-reproduction is certainly one of the most important properties of biological cells (organisms). In previous work[1], the central role of ribosomes (ribosomal protein complexes) in cellular self-reproduction processes (doubling of cells' biomass) was shown, and a detailed analysis of doubling time limits was carried out for different proto-cells (short description of the term is provided in Supplementary Discussion 4) consisting of combinations of cell components forming ribosome-based self-reproduction systems (SRSs) which are defined as sets of essential cell components with functions directly necessary for cell growth and self-reproduction (including polymerization of macromolecules and synthesis of respective monomers/metabolites) at given growth conditions. The absolute minimal doubling times of SRSs are observed if they contain only ribosomal protein complexes. The introduction of other molecular components to the SRS always prolonged the doubling time of the proto-cells, the effects being different depending on the complexity of the SRS and the specific components added.

However, the analysis described in ref. 1 was limited to the doubling of proto-cells because the models did not include all the necessary cellular processes like the cell cycle and the peculiarities of cells beyond the self-replication of the SRS as in complete bacterial cells. The self-replication of cells cannot be considered and analyzed merely as the doubling of biomass (self-replication of SRS)—it should include also regulated divisions of the cells into daughter cells and other related cellular mechanisms. The integration of the cell cycle into the models of self-replication should enable one

to address also other physiological (regulatory) issues that are not covered by the analysis of the proto-cell models alone.

The prokaryotic cell cycle has been described by various theories or models on the molecular (include relevant cell components in detail) or empirical level (simplified interactions between parameters of black box models) that differ mainly by the mechanisms of DNA replication initiation, cell division and cell size homeostasis[2–12]. Historically, one of the most acknowledged empirical cell cycle theories based on data of population averages is Cooper-Helmstetter-Donachie (CHD) which links DNA replication initiation to cell mass[2,3,13,14]. Recently, CHD cell cycle theory has fallen out of favor based on analysis of cell size data from single-cell experiments using stochastic models[12,15,16] that support mostly the so-called adder principle[11,17,18] of cell mass growth (named historically as incremental size model[19]).

However, there is currently no consensus in the world on the actual molecular mechanisms coordinating DNA replication, cell division and growth during cell cycle. Various contradicting models involving synthesis/accumulation thresholds of unknown proteins, DnaA, ppGpp or surface/volume ratio or interacting/independent cellular processes[20–26] have been proposed. Moreover, the validity of the key principles of the original CHD theory has been also confirmed for some growth conditions in single-cell experiments in addition to population experiments with perturbations[27–29]. A number of theoretical models involving the control of DNA replication

[1]Center of Food and Fermentation Technologies, Mäealuse 2/4, 12618 Tallinn, Estonia. [2]Department of Chemistry and Biotechnology, School of Science, Tallinn University of Technology, Akadeemia tee 15, 12618 Tallinn, Estonia. [3]These authors contributed equally: Kristo Abner, Raivo Vilu. ✉e-mail: raivo@tftak.eu

initiation or cell size based on elements of CHD theory have also been constructed recently[27,30].

One of the central issues in CHD theory is the role and determination of unit cell mass ($M_u$), historically also called the initiation mass. $M_u$ is defined as the (constant) ratio of cell mass at DNA replication initiation to the number of replication origins present in the cell at the time of initiation[31]. It can be deduced from this definition that $M_u$ is also the total mass of the slowly growing cell at a certain cell age at which the replication of the single genome is initiated during the cell cycle.

Because the number of replication origins in bacterial cells increases exponentially with a decrease in the cell cycle length (i.e. with an increase in the specific growth rate of the cell culture) according to a discrete mathematical function (assuming overlapping parallel replication processes), DNA replication can not be started at every integer multiple value of $M_u$ but at $2^i$ integer multiples of $M_u$. Based on that a series of formulas have been derived for exponentially growing cell populations as well as for single cells taking into account various cell mass growth laws (linear, exponential)[2,3,13,14,31–37]. $M_u$ is an important input parameter in CHD theory because $M_u$ values determine besides the initiation conditions of DNA replication also the cell mass for all cell cycle length and cell age values. In that sense, $M_u$ is the logical starting point for the quantitative description of the cell cycle, and also for the bottom-up bacterial cell design based on CHD.

Despite the importance, $M_u$ has been overwhelmingly treated as an empirical input parameter whose value is determined preferably experimentally[36,38–40]. As CHD theory describes only DNA replication and cell size growth which are certainly not the entire cell, CHD theory does not provide any specific considerations for the deterministic calculation of $M_u$. Therefore, it would be very useful to combine CHD with the aforementioned models of SRS, because they are perfectly complementary to each other: using them together allows to analyze cells much more thoroughly than using both separately.

There is also a third part that should be added to these two, because besides the SRS and the cell cycle regulation mechanisms, cells obviously contain also some other parts. Those other parts (combined) are designated as cell load (short description of the term is provided in Supplementary Discussion 4) in the current work. The analysis of cell load has been mostly limited to specific protein fraction(s) in the literature[30,41,42]. It has been shown[43,44] that the synthesis of cell load prolongs cell cycle length. However, cell load is also a subsystem of the metabolic machinery of cells which is not directly engaged in self-reproduction and is responsible for carrying out special functions, e.g., containing components responsible for the homeostasis of intracellular pH, components needed for switching between different growth conditions, etc.[45]. Therefore, cell load could include besides proteins also RNAs etc., and cell load could also be a useful load for the SRS in the cell because it provides opportunities to make the cells more flexible or more specific (although slower) depending on the stability of growth conditions.

In principle, cell load could be removed without impairing the logical structure of cell's self-reproduction processes. This means that cell load could also be considered a useful load in a different sense—useful for biotechnological purposes: it can be (theoretically) fully or (in practice) partially replaced with biosynthetic processes and products for the biotechnology industry. Therefore, indirectly, the properties of cell load can also determine the productivity of biotechnologically useful (possibly human-designed) biosynthetic processes in cells.

Thus, our goal in this paper is to solve the following main problems associated with the principles of cell design:

(1) Expanding the self-replication analysis of cell components and proto-cells to complete cells by integrating the synthesis of cell components and CHD cell cycle theory. This would replace the doubling of only biomass with the self-reproduction and division of cells. One of the specific scientific questions that can be studied by such an approach is the question about slow growth of large cells. Our previous analysis showed that the doubling of SRS is in most cases extremely fast with a narrow range of doubling time values except for very small-sized SRSs,

whereas we know that real biological larger cells are able to grow with a very wide growth range.

(2) Expanding the analysis of empirical $M_u$ (central piece in CHD theory) to different unit cell parameters. This would enable to get deterministic $M_u$ (and other unit cell parameters) and provide an opportunity to link it with actual molecular mechanisms behind $M_u$. One of the specific questions that can be studied by such an approach is the question about the growth boundaries of unit cells. CHD theory provides strict dependencies between growth rate and cell size based on the values of $M_u$ and cell cycle periods, but CHD theory does not provide any rules for the determination of (minimal/maximal) $M_u$ values.

(3) The analysis of cell load of unit cells. This would enable to study the effect of cell load on the quantitative properties (including the maximal growth rate) of unit cells and give the theoretical basis for possible biotechnological applications. One of the specific questions that can be studied by such an approach is the question about the optimal region of cell load synthesis productivity in unit cells. While it is already common knowledge that the synthesis of other proteins (including recombinant proteins) decreases the growth rate, the relation between synthesis productivity and cell size or growth rate has not been established.

Simplified stoichiometric models were developed in the present study that can be used for the description and analysis of the growth of unit cells (UCs, cells with the mass equal to $M_u$). CHD cell cycle theory was selected as a basis for the integration of the cell cycle because such an empirical model suited well to our simplified single-cell models of proto-cells that describe the doubling of SRS components (including polymerases, enzymes, RNA, metabolic pathways and cell membrane) based on stoichiometric balances of molecule numbers, mass, time, etc.[1]. The resulting simplified single-cell models of unit cells (SSUCMs) include the same SRSs as in selected simplified single-cell models of proto-cells (SSPCM-SRS-M, SSPCM-SRS-R), but SSUCMs include also cell load (synthesis of generic protein not participating in self-reproduction and accumulating in the cell, named as cushioning protein (CP)) in the case of selected models (SSUCM-M, SSUCM-R).

The developed SSUCMs allowed us to calculate the size, composition and other physiological parameters of UCs. The aim of the present paper is to show the relationships between the $M_u$ and other important physiological parameters to illustrate and prove the claim that $M_u$ is an important parameter in ab initio cell design of bacteria. The analysis indicated that besides possible secondary functions, cell load has also a cushioning role (short descriptions of related terms are provided in Supplementary Discussion 4) allowing to increase the size of the cells and contents of the cellular components also at slow growth. In addition to cell cycle mechanisms involving $M_u$ in DNA replication initiation, an especially important issue is the relationships between $M_u$ and the productivity of cellular biosynthetic processes, which is determined by the peculiarities of linkage of cell load and SRS.

In conclusion, the following main results were obtained:
(1) Simplified stoichiometric cell models were developed by integrating models describing the self-reproduction of proto-cells and the unit cell concept from CHD cell cycle theory.
(2) It was shown that cell load (cushioning protein) secures reasonable quantitative properties (cell size, slow growth) of unit cells.
(3) The models allowed to determine and illustrate the main relations between the size and complexity of SRS, cell load and cell cycle parameters. Also, dependencies of the productivity of cushioning protein synthesis were explained.

## Results

The general motivation of this and previous work[1] came from the principles of bottom-up cell design—how the quantitative properties of individual cell components and intracellular processes affect the growth and properties of cells. The main purpose of the previous work was to improve the theoretical understanding of design principles concerning SRS (including theoretical

doubling time limits and the role of ribosomes), whereas in this paper the objective is to describe and analyze complete cell (unit cells—SRS integrated with cell cycle, including also cell load) in the framework of the UC concept. As already mentioned before, UC parameters ($M_u$) determine DNA replication initiation according to the CHD theory[31]. Therefore, the introduction of complete CHD cell cycle theory requires the description of UC which is deterministic and mechanistic in the current study and not statistical and empirical as usually.

Four different models were constructed and used for the analysis in order to cover the four main possible combinations of the complexity of SRS (small vs. large complexity characterizing rich vs. minimal media growth) and the availability of cell load (CP is synthesized vs. not synthesized). These models are stoichiometric assuming steady-state condition and describe the doubling of the simplified cells and all of the cellular components (proteins, RNA, membrane lipids, DNA) taking into account cellular balances (molecule numbers, time, mass, etc.). A brief overview of how the most important model parameters affect each other is provided in Supplementary Discussion 5.1, but the rigorous mathematical definitions of all the relations in the models can be seen in the code of the used models (Supplementary Software) and are thoroughly described in Supplementary Discussion 5.10.

Although we illustrate the general principles and capabilities of these models with some calculations based on different published experimental data, it was not our goal to find the simplest models to fit the available datasets. Furthermore, most of the available data points are obtained at higher growth rates, whereas unit cells, which are the focus of this paper, are characterized by relatively low growth rates. Proper reproduction of such data points requires extending the models beyond the unit cell condition and, in some cases, making the models strain-specific. This is indeed also possible with our modeling framework, but in this paper we are focusing on establishing the foundations of that modeling framework and specifically develop a theoretical understanding of the unit cell, which we consider a particularly important concept for cell design.

The designations used in this paper and in the model equations are explained in Supplementary Discussions 1-4 (abbreviations, symbols of parameters, terms written in italics and characterizing conformance between the parameter values of our models and living *E. coli* cells, short descriptions of the main concepts including proto-cell, cell load and cushioning).

## Cell cycle of unit cells

CHD cell cycle theory has been one of the most popular models of bacterial cell cycle due to the simplicity of its concepts, although it has been shown that CHD does not describe certain actual experimental conditions, as stated above. CHD comprises several cellular processes: (1) replication of genomic DNA, (2) division of the self-reproduced mother cell into daughter cells, and (3) self-replication (doubling or growth) of the cell. It is assumed in CHD that DNA replication and cell division are mutually coordinated and synchronized processes. Both processes are labeled as periodic processes (clear initiation and termination, take place during certain period of cell cycle), whereas overall cell growth (doubling of cells' biomass) is a continuous process (throughout cell cycle). Although continuous DNA replication might be observed in the case of faster-growing cells with overlapping replication rounds, periodic process refers to the duration of individual chromosome replications.

The cell size, the coordinated patterns of DNA replication and the doubling of cells are determined in CHD cell cycle theory by four parameters ($M_u$, cell cycle length, genome replication time ($t_C$), and division time ($t_D$) as shown, for example, in ref. [31] which we have introduced also into SSUCMs. The physical meanings, reflected molecular mechanisms in cells, determining factors and values of parameters are described in Fig. 1.

The base assumptions (section "Description of models") of the stoichiometric models used in the current work were deliberately selected to describe the cellular parameters of single cells at the cell age at the beginning of the cell cycle ($t_0$) for mathematical reasons. Therefore, it is also convenient

that $M_u$ will equal the mass of a single cell at the beginning of the cell cycle, and to achieve that, we have to select a very specific cell cycle length with which the replication of single genome is initiated also at the beginning of the cell cycle. We can define this specific cell cycle length as the cell cycle length of the unit cell ($t_{CD}$) which is equal to the sum of both cell cycle period parameters ($t_C$, $t_D$):

$$t_{CD} = t_C + t_D \qquad (1)$$

It must be stressed that $t_{CD}$ is not a mathematical shorthand to represent the sum $t_C + t_D$, it is the cell cycle length of the cell that just happens to equal $t_C + t_D$ according to CHD, and as a cell cycle length it also acts as the main time constraint for cell doubling and cellular synthesis processes. Cells that at the beginning of their cell cycle have mass $M_u$ and have a single complete copy of the genome start the replication of their genome also immediately at the beginning of their cell cycle, and the cell cycle is terminated at $t_{CD}$ with the completion of cell division—cell with mass $2M_u$ divides into two cells, each with mass $M_u$.

The value of *approximate* $t_{CD}$ (3520 s = 1.0 h in the case of *E. coli* based on Eq. (1) or growth rate ($\mu$) equal to 0.7 h$^{-1}$ calculated from Supplementary Eq. (115) of ref. 1, $\mu = \ln(2) \cdot 3600/t_{CD}$) is the shortest value of cell cycle length for the cells with a complete single copy of the genome. Indeed, if cells are growing slower then the $M_u$ value is not reached at the beginning of the cell cycle but later and therefore DNA replication starts also later (after passing the prokaryotic cell cycle B phase—see Fig. 1).

This peculiar cell state (corresponding to $M_u$ and $t_{CD}$) is defined as UC in the current work. The parameters of UC (including $M_u$) characterize cells capable of maintaining the sustainable existence of a complete single copy of the genome at growth with $t_{CD}$. Whereas CHD theory provides very strict estimations for cell mass values at particular cell cycle length and $M_u$ values[31], it is silent about the determining rules for $M_u$. Despite the importance, $M_u$ has been handled in the literature as a free variable whose values can be defined freely and estimated from experimental cell mass and replication origin measurements. Certainly, the cell cycle based on CHD theory can be integrated into the SRS analysis framework by assuming that $M_u$ is just an abstract free variable that determines all the other cell parameter values via cell mass. However, it is much more reasonable and insightful for the purposes of bottom-up cell design to first start with modeling and analyzing the parameters of UC.

As mentioned before, we developed simplified single-cell models of proto-cells and used them in the systematic analysis of self-replication processes assuming the central role and mechanisms of ribosomes or ribosomal protein complexes in the SRSs[1]. Ribosomal protein complexes are the only cell components that are able to self-reproduce themselves, whereas all the other cell components are not and inclusion of them always prolongs the doubling time. Therefore, the absolute minimal doubling time (the fastest growth) is determined by the time the protein part of ribosomes can self-reproduce in a simplified approach (equal to the ratio of the number of amino acid molecules in the ribosomal protein complex ($n_{rpc}$) to the apparent working rate of the ribosome ($k_{rs}$), $n_{rpc}/k_{rs}$ and it does not depend on the number of complexes. According to the simplest description, the fraction of ribosomal proteins in the total protein content decreases linearly with growth rate decrease and does not depend on other parameters beside $n_{rpc}$, $k_{rs}$ and $t_{CD}$.

We observed practically constant doubling times of the self-reproduction systems of the abstract proto-cells in a remarkably wide range of values of the number of ribosomes in the cell. The minimal value of the doubling time of SSPCM-SRS-M was 2474 s = 0.7 h and the minimal value of the doubling time of SSPCM-SRS-R was 936 s = 0.3 h as seen from the Figs. 1 and 2 of ref. 1 and Table 1 of ref. 1. These results show that in the case of the nearly constant doubling times of proto-cells, the cell mass values could be changed freely in a wide range. However, proto-cells with physiologically meaningful sizes and intracellular concentrations growing on rich medium were found only in a very narrow range (936–940 s = 0.3 h) of doubling time right next to the growth limit. If proto-cells grew slower

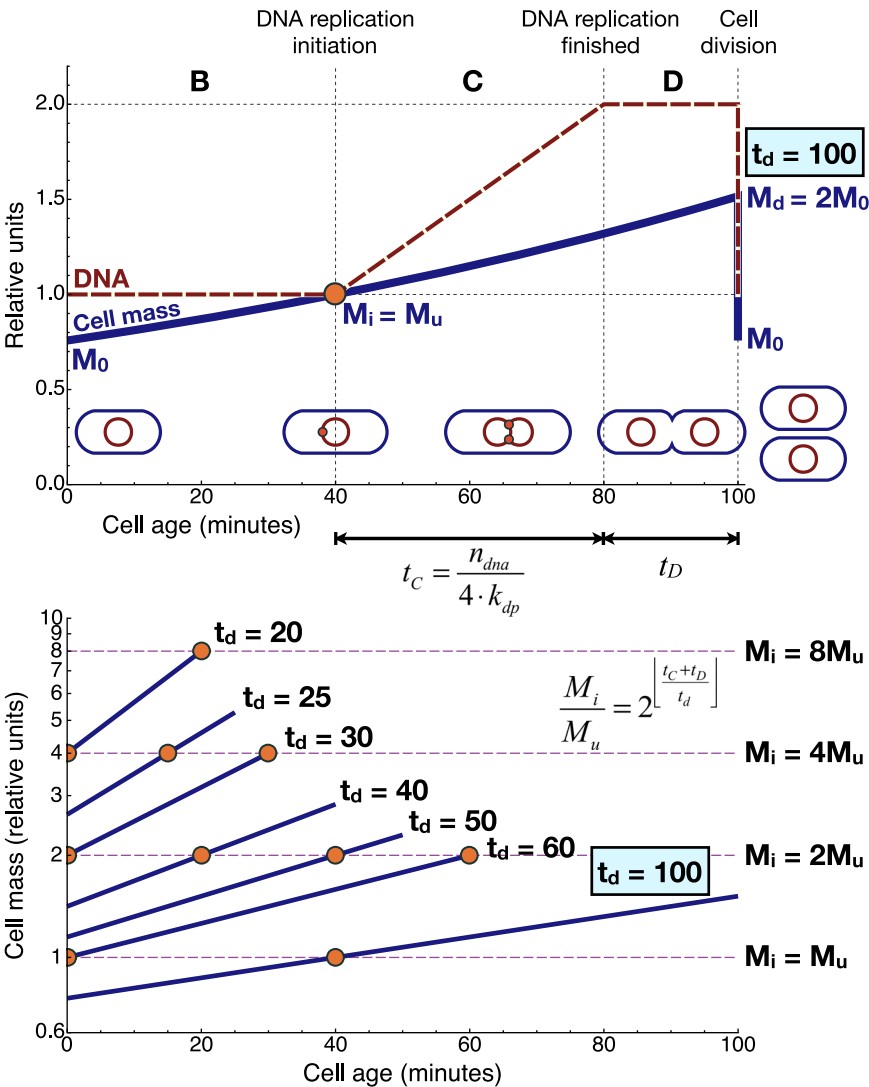

**a) Slow growth at cell cycle length $t_d$ = 100 minutes.**

**b) Examples of growth at various cell cycle lengths $t_d$.**

**Fig. 1 | Overview of Cooper-Helmstetter-Donachie cell cycle theory. a** DNA replication and growth of a slowly growing cell during cell cycle. $M_0$—cell mass immediately after previous cell division, $M_i$—initiation mass, $M_d$—cell mass immediately before division, $M_u$—unit cell mass, $t_d$—cell cycle length, $t_C$—DNA replication time, $t_D$—division time, $n_{dna}$—the number of deoxyribonucleotide molecules in the genome, $k_{dp}$—the apparent working rate of DNA polymerase. DNA replication starts after achieving $M_i$ and is terminated after the completion of a new genome (C phase in CHD). Cell division processes are initiated immediately after the termination of DNA replication and are terminated with the separation of both cells at the end of the cell cycle (D phase in CHD). In a slowly growing cell there exists also period B (here 40 min) before DNA replication. **b** Relationships between cell mass and cell age in a semilogarithmic graph at different $t_d$ values. The varying values of DNA replication initiation age are derived at constant $t_C$ and $t_D$. The value of $t_C$ incorporates a number of different processes of DNA replication initiation, genome replication and chromosome segregation[67]. Because genome replication and

segregation occur simultaneously in prokaryotes, $t_C$ is not the sum of periods of individual processes. Although little is known about the effect of genome segregation on the overall timing of DNA replication, it is reasonable to assume that $t_C$ is determined primarily by the speed of the replication fork. The latter is mostly determined by the speed of DNA polymerase III according to the literature[68]. In most of our calculations $t_C = 2320$ s = 0.6 h (Supplementary Eq. (114) and Supplementary Table 24 of ref. 1). $t_D$ is determined by the duration of various molecular processes (divisome formation, functioning of the MinCDE system and nucleoid occlusion, Ter linkage, transport of chromosomes, septum formation) but very little is known about their timing[36] and effect on the value of $t_D$. In most of our calculations $t_D = 1200$ s = 0.3 h (ref. 13). The molecular/regulatory mechanisms constraining the values of $M_u$ are not known. The range of $M_u$ is approximately $1·10^{-13}$–$1·10^{-12}$ g (unit cell)$^{-1}$ (refs. 28,69). $M_u$ is also the mass of such cells that contain only a single replication origin and only a single genome is replicated during the cell cycle.

(comparable to the regular cell cycle length values of actual living cells), then the respective proto-cells were extremely small (numbers of cell component molecules/complexes in the cell <1 molecule (cell comp) cell$^{-1}$) indicating that the doubling of SRS is clearly not sufficient to explain the mechanisms of slow growth of large cells.

At the same time the remarkable differences in the minimal values of doubling times of proto-cells showed unequivocally that the cell mass values

depend on the complexity of SRS in the case of cell components that are stoichiometrically fixed to ribosomes. These conclusions are, in fact, in agreement with the remark made above of $M_u$ considered as a free parameter, at least if the numbers of cell component molecules/complexes in the cell are also changed together with $M_u$ (assuming that the mathematical properties of simplified single-cell models of proto-cells and SSUCMs are similar because the pair of SRSs (corresponding to minimal and rich media)

used in both cases is identical). These considerations are linking the values of $t_{CD}$, $M_u$ and numbers of cell component molecules/complexes in the cell with each other by taking into account the patterns of metabolic fluxes during the doubling of the cellular mass, DNA replication and cell division in the models developed. The analysis of these patterns and coordination with the cell cycle is one of the main subjects of the present paper.

To sum up, the CHD cell cycle theory is very suitable for integration with our proto-cell models. In the next section we take a closer look at the properties of the resulting combined models.

## Calculated unit cells based only on self-reproduction systems are problematic

First, we calculated the $M_u$ and other parameters of UC at *approximate* $t_{CD}$ = 3520 s = 1.0 h using SSUCM-SRS-M and SSUCM-SRS-R, which are just SRSs at unit cell condition (different calculations have been described shortly in section "Description of models"). The results of the calculations (Supplementary Tables 13 and 14) showed that the UC of SSUCM-SRS-M was considerably larger than the UC of SSUCM-SRS-R, reflecting the differences between the complexities of the SRSs. Interestingly, similar differences between $M_u$ values were observed also in the reanalyzed experimental data of *E. coli* cells growing on different media[46] (Supplementary Discussion 5.2). These results are in accordance with metabolic scaling studies of prokaryotes indicating that cell size increases linearly with the size of the metabolic network[47]. The analysis based on ref. 46 did not take into consideration the well-known differences in $t_{CD}$ values on different media, although the variations have been reported to be very small between certain rich and minimal media[48]. On the other hand, studies based on other *E. coli* strains and taking into account $t_{CD}$ differences have been published that exclude the dependence between $M_u$ and growth conditions almost completely[28,49].

The values of $M_u$ (or initiation mass equal to 1 $M_u$) estimated from experimentally determined data (cell size, cell cycle periods, growth rate) of exponential cultures of *E. coli* in different growth conditions[28,36,46,49] are in the range of $1 \cdot 10^{-13}$–$1 \cdot 10^{-12}$ g (unit cell)$^{-1}$. The numbers of ribosomes in the cell estimated from experimentally determined data[36] depend on the growth rate and are in the range of $8 \cdot 10^3$–$7 \cdot 10^4$ ribosomes cell$^{-1}$ (respective range of molar concentrations of ribosomes is $2 \cdot 10^{-4}$–$4 \cdot 10^{-4}$ mol L$^{-1}$). As seen from Table 1, the values of calculated $M_u$ = $3.77 \cdot 10^{-14}$ g (unit cell)$^{-1}$ (SSUCM-SRS-M) and $1.91 \cdot 10^{-14}$ g (unit cell)$^{-1}$ (SSUCM-SRS-R) are smaller than the expected range of $1 \cdot 10^{-13}$ – $1 \cdot 10^{-12}$ g (unit cell)$^{-1}$ and the reanalyzed published data (Supplementary Discussion 5.2). In addition, the numbers of ribosomes in the cell are very low (13 ribosomes cell$^{-1}$ in SSUCM-SRS-R and 337 ribosomes cell$^{-1}$ in SSUCM-SRS-M, $1.1 \cdot 10^{-6}$ and $1.5 \cdot 10^{-5}$ mol L$^{-1}$) and the total dry weight content of DNA fraction in the cell is anomalously high —82.7% and 42.0% respectively. Moreover, even in the case of minimal $t_{CD}$, the corresponding values of $M_u$ would still be considerably less than $5 \cdot 10^{-12}$ g (unit cell)$^{-1}$ (Supplementary Tables 13 and 14). The value of minimal $t_{CD}$ was 2474 s = 0.7 h for SSUCM-SRS-M and 936 s = 0.3 h for SSUCM-SRS-R. It is reasonable to explain the abnormalities in cell parameters with the very low numbers and content of ribosomes in the cell. It is not possible to design physiologically "normal", viable cells if there is no possibility for having sufficient amounts of ribosomes. Basically, these results were not unexpected considering the observed possible difficulties of slow growth of proto-cells on the rich medium described in the previous section.

However, at the same time it is not possible to state that the calculated $M_u$ values were below minimal theoretical cell sizes because similar-sized or even smaller cells[50–52] do exist that are also characterized by high content of DNA and low numbers of ribosomes in the cell (Supplementary Discussion 5.3). Theoretical metabolic scaling studies have also indicated that minimal cells should be mostly filled with DNA[53]. On the contrary to current calculations, most of the experimentally observed or theoretically obtained values of cell mass and numbers of ribosomes in the cell comparable to values in Table 1 correspond to very slow growth and a positive correlation between cell size and growth rate has been proposed[47].

## Table 1 | Unit cells growing on rich and minimal media

| Model | $M_u$ | $N_{rs}$ | $C_{rs}$ | DNA%$_{mmc}$ |
|---|---|---|---|---|
| SSUCM-SRS-R | $1.91 \cdot 10^{-14}$ | 13 | $1.1 \cdot 10^{-6}$ | 82.7 |
| SSUCM-SRS-M | $3.77 \cdot 10^{-14}$ | 337 | $1.5 \cdot 10^{-5}$ | 42.0 |

Calculated values of unit cell mass ($M_u$, g (unit cell)$^{-1}$), the number of ribosomes in the cell ($N_{rs}$, molecules (rs) cell$^{-1}$), the molar concentration of ribosomes ($C_{rs}$, mol (rs) L$^{-1}$) and total dry weight content of DNA fraction in the cell (DNA%$_{mmc}$, % (g (tot dna) (g (dw cell))$^{-1}$)) of unit cells growing on rich (SSUCM-SRS-R) and minimal medium (SSUCM-SRS-M) if the cell cycle length of the unit cell $t_{CD}$ is equal to *approximate* 3520 s = 1.0 h.

To sum up, UCs that are composed of only SRS components have difficulties with slow growth. The next section explains how the difficulties with slow growth appear to be solved in real biological cells.

## Introducing the cushioning protein

The most straightforward option to resolve these problems of the size and composition of UCs is to include the synthesis of cell load and to introduce (an abstract?) cushioning parameter which is not directly necessary for self-replication but allows to reach a physiologically reasonable composition and size of the UC at slower growths. Indirectly, the cushioning parameter represents the united functioning of cell load components (such as reserve substances, part of regulatory proteins) and also unused SRS components (such as inactive ribosomes, enzymes of inactive pathways). A longer discussion about the considerations and necessity of cushioning parameter in UC is provided in Supplementary Discussion 5.4.

Currently, cushioning parameter and cell load are represented in our models by the cushioning protein (CP). CP is synthesized and accumulated in UCs and it does not have any specific function in the models other than providing cushioning. However, the introduced CP should be interpreted as a set of different proteins with various functions in real cells as in ref. 42 where analogous unspecified protein with average amino acid composition representing structural, signaling, household and unused metabolic proteins was used to model the peculiarities of yeast growth. Compared to the proteins of SRS which must assure the self-replication of cells, CP can have very different arbitrary functions and molecular properties. Another very important aspect of the introduction of CP into the cells is that it might also represent the useful load (for example recombinant proteins synthesized by the cell factories), which is of high practical interest in biotechnology[54].

Separating the cellular metabolic machinery into these two subsystems (SRS, cell load) is not in all cases unequivocal and simple mainly due to the ambiguous position of unused SRS cell components. Quantitatively, the same cell component can be divided into different sub-populations considering different cell processes (for example ribosomes that polymerize themselves, polymerases or CP). It means that the same cell component might belong to both groups, SRS and cell load. Ribosomes that are synthesizing only CP are not strictly required for self-reproduction (Supplementary Discussion 5.5). Also, the precise quantitative division for specific strains is possible only based on the analysis of experimental results[55]. This is why in this paper we are analyzing simplified models of cells with a daisy-like structure of the metabolic network consisting of independent linear reaction chains responsible each for the synthesis of an individual biopolymer or building blocks for the synthesis pathways. Dividing the metabolism into SRS and cell load is more intuitive and simpler in these models and thus allows to demonstrate and explain the fundamental principles more clearly.

To sum up, biological cells seem to contain what can be called cushioning parameters (which can be physically realized in a number of different ways) and therefore we also added cushioning to our models. In the next section we show how the inclusion of CP improves the UCs.

## Cushioning protein ensures physiologically reasonable cells

Let us now see how the cell composition and in fact all the physiological parameters could be controlled (changed) in the UC of *E. coli* by introducing the CP into the cell and adjusting the number of ribosomes in the cell and the number of cushioning protein molecules in the cell ($N_{cp}$) based on SSUCM-

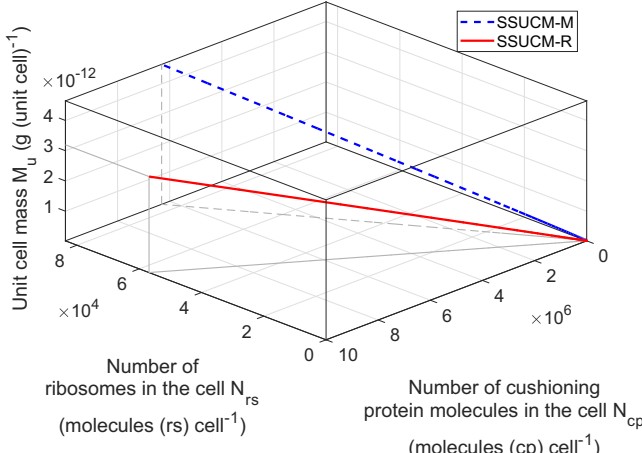

**Fig. 2 | Cushioning protein synthesis in unit cells (UCs).** The relationship between the number of cushioning protein molecules in the cell ($N_{cp}$, molecules (cp) cell$^{-1}$), the number of ribosomes in the cell ($N_{rs}$, molecules (rs) cell$^{-1}$) and unit cell mass ($M_u$, g (unit cell)$^{-1}$) of UCs growing on rich (solid line) and minimal medium (dashed line) if the cell cycle length of the unit cell $t_{CD}$ is equal to *approximate* 3520 s = 1.0 h. The respective dependencies for molar concentrations are visualized in Supplementary Fig. 5.

M and SSUCM-R. Figure 2 illustrates the possibility to adjust $M_u$ by changing the number of ribosomes in the cell (characterizes the size of SRS) and $N_{cp}$ (characterizes the size of cell load) in the case of *approximate* $t_{CD}$ = 3520 s = 1.0 h.

It must be stressed that the growth point at $N_{cp} = 0$ in Fig. 2 is actually the growth boundary point because it is not possible to decrease the value of $M_u$ any further by decreasing the value of $N_{cp}$. This is the lower size limit of UCs. In addition, there exist also upper size limits of UCs (not visible in Fig. 2) but those growth boundaries are determined by the cell membrane area due to the shrinking of the fraction of the membrane surface area covered by membrane lipids. The latter is caused by the change of surface to volume ratio of the cell and the need for more membrane proteins in larger cells.

It can be seen in Fig. 2 that to reach the same value of $M_u$ we need to introduce more CP to the SSUCM-R than to the SSUCM-M cells because the size of SRS of SSUCM-R is smaller than the size of SRS of SSUCM-M (i.e., $M_u$ of SSUCM-R < $M_u$ of SSUCM-M at the same $N_{cp}$ value). The characteristic linear dependencies between different parameters can be explained by the constant values of the apparent working rates of catalyzing cell components, which means that the rates of synthesis processes are regulated via the numbers of cell component molecules/complexes in the cell. The increase of CP synthesis also requires a proportional increase of the number of ribosomes and other synthesis equipment in the UC which leads to the proportional increase of $M_u$.

Indirect evidence for cushioning mass in UC can be also found from the reanalysis of the data of different *E. coli* strains grown on various mineral and rich media[28,46,49] using SSUCM-R for complex and SSUCM-M for minimal growth media (Supplementary Discussion 5.6). The calculations indicate that cells growing on complex media should contain more CP assuming that their SRS size is smaller compared to cells with similar size and growing on minimal media. Such differing ratios of cell load to SRS size between growth media types explain also the similar $M_u$ values mentioned in the previous section. The dependence between the total mass of cushioning protein molecules in the cell and $M_u$ was characteristically linear as in Fig. 2 in the case of calculations based on *approximate* $t_{CD}$ but such clear dependencies were lost if experimentally determined varying $t_{CD}$ values were used. However, those calculation results should be taken into account very carefully for the characterization of different strains because the correspondence between the models and physiological conditions in the experiments might be insufficient

(especially in the case of semi-complex media with only part of the monomers supplemented).

Some additional dependencies of the most important UC parameters are described in Supplementary Discussion 5.7 due to limited space but an exhaustive overview will be provided in a future publication.

To summarize: the introduction of CP into the previously described somewhat "abnormal" cells (Table 1) allows these cells to reach reasonable cellular parameter values (including $M_u$) and ensures the viability of the model cells. And, importantly, these models make it possible to quantitatively describe UCs in a level of detail that is impossible in the classical CHD theory. In the next section we show that the properties of UCs depend also on CHD parameters.

### Changing $t_{CD}$ has major effects on properties of unit cells

The dependencies of the size and composition of UC are determined by the size and complexity of SRS and by the total mass of CP accumulated into the UC as explained above (Fig. 2). Another important parameter that constrains the parameter values of UC is $t_{CD}$. The varying of $t_{CD}$ has potentially also practical value for biotechnology because the molecular mechanisms controlling the characteristics of cell cycle events ($t_D$ and the apparent working rate of DNA polymerase) can be modified more easily than, for example, the sizes of SRS and cell load. The dependencies between the main parameters of UC and $t_{CD}$ (Figs. 3–5) are described below and they could be used to develop cases of practical guidance in designing and adjusting the parameters of UCs.

Firstly, the growth boundaries of UCs (minimal $t_{CD}$) are characterized by lines in the case of varying $t_{CD}$ (Figs. 3–5) instead of points as in Fig. 2. The selected range of $t_{CD}$ values in this paper is larger than the range of experimentally observed values of *E. coli*[36,48]. The lower $t_{CD}$ values (<3·10$^3$ s = 0.8 h) were included in the analysis in order to calculate and visualize theoretical growth boundaries.

The minimal $t_{CD}$ values of SSUCM-M and of the smaller UCs of SSUCM-R correspond to the case of $N_{cp} = 0$ (Fig. 4), meaning that the cells are fully occupied by the SRS components including those proteins that are necessary for self-reproduction. A similar but not identical type of growth boundary constraints has been proposed already earlier[42].

The values of minimal $t_{CD}$ are differing between models. The SRS of SSUCM-R is smaller (can be deduced from Fig. 5c) and the minimal $t_{CD}$ value is lower than in the case of SSUCM-M. As already reported earlier[1], faster growth is possible with smaller SRS complexities because the content of ribosomes is higher in these cases and the doubling time is nearer to the value of the *approximate* doubling time of a single ribosomal protein complex which is the absolute theoretical growth limit of cells. This also explains the differences in minimal $t_{CD}$ values of the present models and it is important in the further analysis of the productivity of cells in biosynthetic processes (in the next section). The possibility to calculate the growth boundary is an important feature of the SSUCM framework for the designers of cell factories.

It can be seen in Fig. 3 that the values of $M_u$ of SSUCM-M and SSUCM-R are almost the same at a given $t_{CD}$ and at the same number of ribosomes in the cell although the values of the mass of the SRS of the cell and the total mass of CP molecules in the cell are remarkably different.

The reason for the nearly constant ratio of $M_u$ to the number of ribosomes in the cell (nearly constant molar concentration of ribosomes) between models is that cell load is intracellular protein. Ribosomes are synthesizing the majority of the SRS size (all necessary proteins) but also CP. If the complexity of SRS is larger (more SRS component species), then cells have less CP because a bigger fraction of ribosomes is allocated to the self-replication of SRS. In the cases of SSUCM-M and SSUCM-R this means that the enzymes of metabolic pathways are replaced by CP or vice versa. The sum of the mass of the SRS and the total mass of cushioning protein molecules in the cell is equal to $M_u$. If the cell load would have been something else (for example glycogen), then the values of $M_u$ of SSUCM-M and SSUCM-R would have been different at the same number of ribosomes in the cell because then the proteins would have not been merely replaced by other proteins.

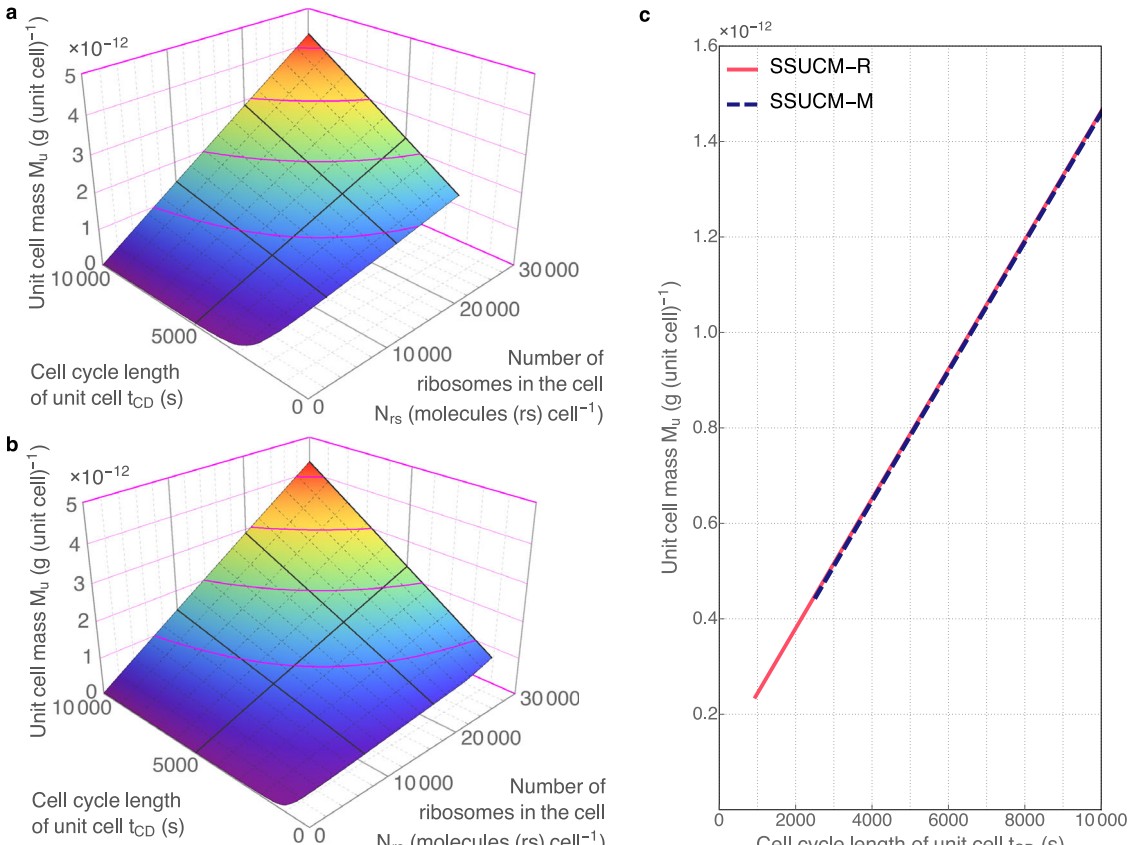

**Fig. 3 | Unit cells growing on minimal and rich media. a** The relationship between unit cell mass ($M_u$, g (unit cell)$^{-1}$), cell cycle length of the unit cell ($t_{CD}$, s) and the number of ribosomes in the cell ($N_{rs}$, molecules (rs) cell$^{-1}$). Calculations and regression surfaces were obtained for various $N_{rs}$ (<3·10$^4$ molecules (rs) cell$^{-1}$) and $t_{CD}$ values (<10$^4$ s = 2.8 h) using SSUCM-M. The visualized range of $N_{rs}$ corresponds to the range of molar concentration of ribosomes $C_{rs}$ < 3.9·10$^{-5}$ mol (rs) L$^{-1}$. **b** Similar

relationship in the case of SSUCM-R. The visualized range of $N_{rs}$ corresponds to the range of $C_{rs}$ < 7.1·10$^{-5}$ mol (rs) L$^{-1}$. **c** The relationship between $M_u$ and $t_{CD}$ in the cases of rich (solid line) and minimal medium (dashed line) at $N_{rs}$ = 10$^4$ molecules (rs) cell$^{-1}$ (equivalent to the range of $C_{rs}$ = 1.1·10$^{-5}$–3.7·10$^{-5}$ mol (rs) L$^{-1}$ for SSUCM-M and the range of $C_{rs}$ = 1.1·10$^{-5}$–7.0·10$^{-5}$ mol (rs) L$^{-1}$ for SSUCM-R).

The reason why the values are almost but not exactly the same in Fig. 3 is determined by the composition of the cell membrane. There are visually almost non-existent differences in the levels of most of the protein fractions of SRS between the models that are caused by the lack of the monomer synthesis energy costs in SSUCM-R. Lower energy costs mean that the requirement for electron transport chain (ETC) complexes is considerably smaller and, therefore, a larger surface area is covered by lipids on the membrane. Altogether, considerable differences in membrane composition lead to small differences in $M_u$ values between the models. Note that the total mass of cushioning protein molecules in the cell decreases more slowly than $M_u$ with $t_{CD}$ decrease meaning that the mass of the SRS of the cell decreases faster (Fig. 5).

The contour lines on the surfaces designate the dependence between the number of ribosomes in the cell and $t_{CD}$ at $M_u$ = const. The dependencies between other cell parameters and $t_{CD}$ have not been visualized here but they are qualitatively very similar to the described surfaces (Figs. 3–5) with varying $M_u$ values except in a few cases. Shortly, when $t_{CD}$ decreases, the numbers of cell component molecules/complexes in the cell belonging to SRS (except lipids) mostly increase, accompanied also by an increase in corresponding synthesis fluxes, by the usual changes of cell composition (an increase in the total dry weight content of the RNA fraction in the cell due to an increase in the number of ribosomes in the cell, a decrease in the total dry weight content of lipid macromolecular fraction in the cell due to the changes in the surface/volume ratio and in the demand for membrane proteins, a decrease in the total dry weight content of the protein fraction in the cell due to a decrease in $N_{cp}$, etc.) and by a gradual cell mass reallocation

to the membrane (an increase in the numbers of ETC complexes and transport protein complexes in the membrane).

The aforementioned few exceptions of cell parameter dependencies are related to DNA and membrane lipids. In the case of $M_u$ = const, the total dry weight content of DNA fraction in the cell is constant as there is only 1 chromosome in the UC, and the number of cell membrane lipid molecules in the cell ($N_{lip}$) decreases continuously when $t_{CD}$ decreases because the surface area of the cell (membrane) is constant and additional membrane proteins are included at the expense of lipids. In the case of varying $M_u$, the total dry weight content of DNA fraction in the cell is obviously not constant and decreases smoothly with the decrease in $t_{CD}$ whereas the dependence of $N_{lip}$ is more complicated and involves an optimum near minimal $t_{CD}$.

To summarize, the developed models allow to describe the theoretical growth of UCs at different values of various cellular parameters (including $t_{CD}$, complexity of SRS and $M_u$). In the next section we show the correspondence between model results and experimentally determined data from the literature.

## Calculations reproduce reasonably the experimentally determined trends in unit cells

To illustrate the capabilities of the developed SSUCM-M and SSUCM-R, they were also used for the calculation of UC parameter values (Supplementary Discussion 5.8) based on experimentally determined cell size, growth rate and cell cycle period data[28,36,46,49] of different *E. coli* strains grown on several complex and minimal growth media. It was shown that calculated cell sizes and numbers of ribosomes of average unit cells aligned well with known experimentally determined trends in the literature.

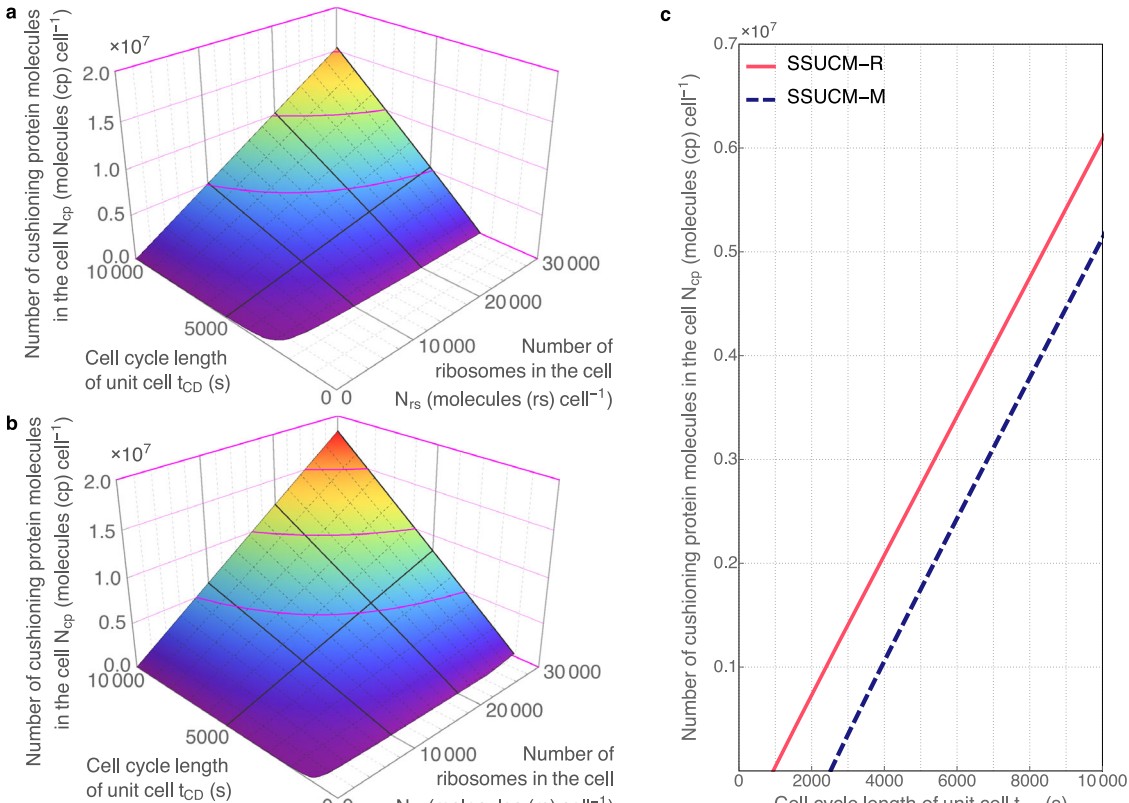

**Fig. 4 | Cushioning protein in unit cells growing on minimal and rich media.** **a** The relationship between the number of cushioning protein molecules in the cell ($N_{cp}$, molecules (cp) cell$^{-1}$), the number of ribosomes in the cell ($N_{rs}$, molecules (rs) cell$^{-1}$) and cell cycle length of the unit cell ($t_{CD}$, s). Calculations and regression surfaces were obtained for various $N_{rs}$ (<3·10$^4$ molecules (rs) cell$^{-1}$) and $t_{CD}$ values (<10$^4$ s = 2.8 h) using SSUCM-M. The visualized range of $N_{rs}$ corresponds to the range of molar concentration of ribosomes $C_{rs}$ < 3.9·10$^{-5}$ mol (rs) L$^{-1}$. The visualized range of $N_{cp}$ corresponds to the range of molar concentration of

cushioning protein $C_{cp}$ < 0.0060 mol (cp) L$^{-1}$. **b** Similar relationship in the case of SSUCM-R. The visualized range of $N_{rs}$ corresponds to the range of $C_{rs}$ < 7.1·10$^{-5}$ mol (rs) L$^{-1}$. The visualized range of $N_{cp}$ corresponds to the range of molar concentration of cushioning protein $C_{cp}$ < 0.0070 mol (cp) L$^{-1}$. **c** The relationship between $N_{cp}$ and $t_{CD}$ in the cases of rich (solid line) and minimal medium (dashed line) at $N_{rs}$ = 10$^4$ molecules (rs) cell$^{-1}$ (equivalent to the range of $C_{rs}$ = 1.1·10$^{-5}$–3.7·10$^{-5}$ mol (rs) L$^{-1}$ for SSUCM-M and the range of $C_{rs}$ = 1.1·10$^{-5}$–7.0·10$^{-5}$ mol (rs) L$^{-1}$ for SSUCM-R).

The dataset of ref. 46 did not include cell cycle periods which is why the parameter values were calculated at *approximate* $t_{CD}$. Therefore, the calculated dependencies between different parameters were characterized mainly by simple linear relationships and significant differences between growth media, matching well with the theoretical results in the previous sections. For example, the calculated ratio of $M_{rna}$ to $M_{prot}$ was constant in all cases and UCs growing on minimal media were bigger.

On the other hand, the data of refs. 28,49 included also experimentally determined $t_{CD}$ values that varied between strains and growth media used. The incorporation of that data into the calculations transformed the simple relationships into more complicated patterns. For example, the calculated ratio of $M_{rna}$ to $M_{prot}$ was not constant but decreased gradually from 0.07 to 0.03 with the increase in $t_{CD}$.

It must be pointed out that the ratio of $M_{rna}$ to $M_{prot}$ was also experimentally determined by ref. 28. However, the cultivation experiments were carried out using turbidostat method (growth points were located presumably at maximal growth rate values) but the models presented in this paper are intentionally limited to UCs as already explained before. Therefore, it was not reasonable to do direct fitting of the models to the measured data as only a few of the experimental data points were measured at slow growth. The comparison of calculated and measured values of the ratio of $M_{rna}$ to $M_{prot}$ showed that in the case of standard input parameter values the calculated values of $M_{rna}$ to $M_{prot}$ were considerably lower than experimentally determined values (approximately 0.15). The reasons for such discrepancy might form a long list—the inability of simplified models to describe precisely the

specific strains (metabolic networks, missing cell wall components), the assumption of cushioning mass being composed of only CP and excluding other possibilities like inactive ribosomes, growth-dependent cell parameter values, etc.

Further calculations with the few almost overlapping slow-growth data points showed that almost perfect fit was possible if, for example, the value of $k_{rs}$ was lowered from 20 to 3–7 molecules (aa) s$^{-1}$ ribosome$^{-1}$. It must be stressed that the standard value of $k_{rs}$ in SSUCMs is probably the possible upper limit which is achieved during very fast growth. Slower growth, however, is characterized by lower $k_{rs}$ values. On the other hand, the comparison of experimentally determined values[36] of the number of ribosomes in the cell and the respective calculated values of UCs from SSUCM-M shows that all the values fitted more or less to the same curve without changes in $k_{rs}$ value (using our standard value $k_{rs}$ = 20). Therefore, there seem to be considerable differences between different datasets in the literature which makes it impossible to always use only a single set of input parameter values for the model. Most probably, the discrepancies mentioned above are actually due to a combination of various factors and strain-specific larger models are needed for clarification. Constructing such larger models is indeed also possible with our modeling framework, but, as already explained before, that was not the aim of this paper.

To sum up, the calculated parameter values of UCs align reasonably well with some experimentally determined data. In the next section we give an example of how such models can provide also insights that have practical relevance to biotechnology.

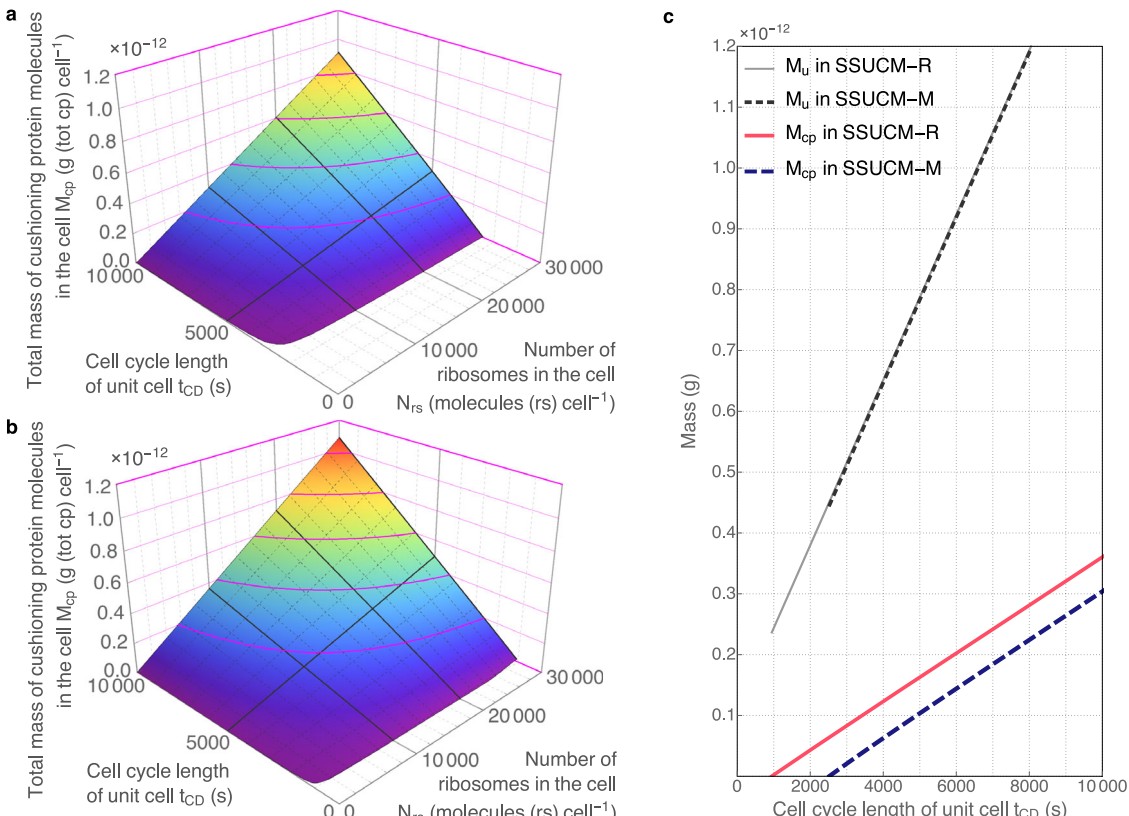

**Fig. 5 | Cushioning mass in unit cells growing on minimal and rich media. a** The relationship between the total mass of cushioning protein molecules in the cell ($M_{cp}$, g (tot cp) cell$^{-1}$), the number of ribosomes in the cell ($N_{rs}$, molecules (rs) cell$^{-1}$) and cell cycle length of the unit cell ($t_{CD}$, s). Calculations and regression surfaces were obtained for various $N_{rs}$ (<$3 \cdot 10^4$ molecules (rs) cell$^{-1}$) and $t_{CD}$ values (<$10^4$ s = 2.8 h) using SSUCM-M. The visualized range of $N_{rs}$ corresponds to the range of molar concentration of ribosomes $C_{rs}$ < $3.9 \cdot 10^{-5}$ mol (rs) L$^{-1}$. **b** Similar relationship in the case of SSUCM-R. The visualized range of $N_{rs}$ corresponds to the range of $C_{rs}$ < $7.1 \cdot 10^{-5}$ mol (rs) L$^{-1}$. **c** The relationship between unit cell mass ($M_u$, g (unit cell)$^{-1}$), $M_{cp}$ and $t_{CD}$ in the cases of rich (solid lines) and minimal medium (dashed lines) at $N_{rs} = 10^4$ molecules (rs) cell$^{-1}$ (equivalent to the range of $C_{rs} = 1.1 \cdot 10^{-5}$–$3.7 \cdot 10^{-5}$ mol (rs) L$^{-1}$ for SSUCM-M and the range of $C_{rs} = 1.1 \cdot 10^{-5}$–$7.0 \cdot 10^{-5}$ mol (rs) L$^{-1}$ for SSUCM-R).

## The productivity of cushioning protein synthesis in unit cells has optima

It should be emphasized that the introduction of the concepts of SRS and cell load allows to analyze the productivity of processes of microbiological synthesis. The load-bearing capacity of the cell (amount of CP) can be characterized by the specific productivity of CP synthesis ($Q_{cp}$) defined by Eq. (2):

$$Q_{cp} = \frac{N_{cp}}{M_u} \cdot \mu \qquad (2)$$

where the growth rate $\mu$ (h$^{-1}$) is expressed by Supplementary Eq. (115) of ref. 1 ($\mu = \ln(2) \cdot 3600/t_{CD}$). Note that in the context of our single-cell modeling framework the most natural way to define $Q_{cp}$ would be the number of CP molecules produced per cell and per second, but as the goal of this section is to demonstrate the practical relevance of the results to the biotechnology industry, we are instead using here the classical industry units based on the doubling of biomass and on hours.

The dependencies between $Q_{cp}$ (can be interpreted as a useful load) and other cell parameters in SSUCM-M and SSUCM-R are presented in Figs. 6 and 7.

As can be seen in Fig. 6, the dependence of $Q_{cp}$ on the number of ribosomes in the cell at fixed *approximate* $t_{CD} = 3520$ s = 1.0 h is hyperbolic (but linear on the molar concentration of ribosomes). The $Q_{cp} = 0$ condition at low number of ribosomes in the cell is explained by the fact that the doubling time of the corresponding SRS is relatively long, and this leaves no space (time) for the synthesis of CP in UC as explained earlier. The hyperbolic dependence appears despite the general proportional (linear)

increase of $N_{cp}$ and $M_u$ in UC and is caused by the mass of the SRS of the cell which comprises a considerable part of $M_u$ in smaller UCs (and is equal to $M_u$ at $Q_{cp} = 0$ and at minimal $t_{CD}$) but is insignificant for larger UCs. This means that the plateau of the dependence is caused by an almost proportional increase of $N_{cp}$ and $M_u$.

The growth boundary of $Q_{cp} = 0$ at minimal $t_{CD}$ values (Fig. 7) is explained by reaching the limit of the doubling time of the SRS in the case of small UCs. In the case of larger UCs, $Q_{cp} > 0$ can be observed at minimal $t_{CD}$ for SSUCM-R (on the right side of Fig. 7b) because there the growth boundary is determined by membrane occupancy as explained earlier.

From the practical (biotechnological) point of view, it is important to understand the peculiarities of the increase in $Q_{cp}$ when increasing the $t_{CD}$. First of all, the most important thing to notice is that $Q_{cp}$ is higher for SSUCM-R in comparison with SSUCM-M. The maximum specific productivity of CP synthesis ($Q_{cp\_max}$) of SSUCM-R is more than two times higher than of SSUCM-M because at the same number of ribosomes in the cell and $t_{CD}$ values the SRS of the former is smaller than the SRS of the latter. Also, the growth range of SSUCM-R is wider and the $Q_{cp}$ dependence has a steeper slope.

As can be seen most clearly in Fig. 7c, $Q_{cp}$ decreases when $t_{CD}$ is increased further into the slower growth region. $N_{cp}$ increases there together with $t_{CD}$ because ribosomes do not have to synthesize so many SRS components (there is more time to double the cell) and the excess capacity is used to synthesize CP. However, the increase in $N_{cp}$ is accompanied by an almost proportional increase in $M_u$, and given that $t_{CD}$ is also increasing and thus growth rate is decreasing, $Q_{cp}$ has to decrease according to Eq. (2).

Therefore, $Q_{cp}$ does not change monotonously when $t_{CD}$ is changed, and there are local maxima for both cells. In the region of fastest growth

there is not much room for CP because SRS takes up most of the cell space and most of the synthesis capacity is needed to replicate the SRS. Meanwhile, in the region of slow growth it is the slowness itself that lowers the

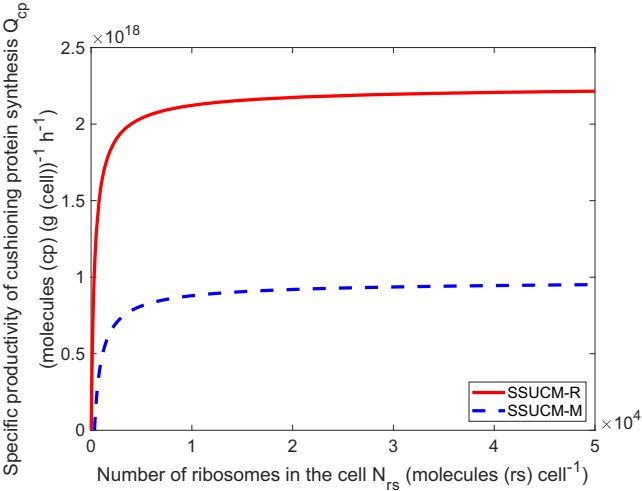

**Fig. 6 | Productivity of unit cells (UCs).** The relationship between the specific productivity of cushioning protein synthesis ($Q_{cp}$, molecules (cp) (g (cell))$^{-1}$ h$^{-1}$) and the number of ribosomes in the cell ($N_{rs}$, molecules (rs) cell$^{-1}$) of UCs growing on rich (solid line) and minimal medium (dashed line) if the cell cycle length of the unit cell $t_{CD}$ is equal to *approximate* 3520 s = 1.0 h. The respective dependencies on the molar concentration of ribosomes (corresponding to the ranges of visualized $N_{rs}$ values) are visualized in Supplementary Fig. 57.

productivity, because productivity is defined per unit of time. The optimum is located between these two regions. The models allow to determine the location of that optimum (i.e. the cell cycle length $t_{CDopt}$ at which $Q_{cp\_max}$ occurs) by numerical calculations, but also through a simplified derivation (Supplementary Discussion 5.9). According to the derivation, $t_{CDopt}$ is approximately twice of that needed for the reproduction of SRS, i.e. approximately twice the minimal $t_{CD}$.

To sum up, the productivity of CP synthesis can be considered to characterize also cell's capacity to synthesize biotechnologically useful products, and, therefore, our modeling framework allows to find optimal parameter regions also for biotechnologically useful synthesis.

## Discussion

In this work, the concept of UC was defined and introduced into previously developed simplified stoichiometric models[1] as the first step for the integration of prokaryotic cell cycle based on CHD theory. Preliminary characterization and analysis of the growth of UCs was carried out based on corresponding models (SSUCMs). In the analysis we focused on global cell parameters (complexity of SRS, size of SRS (characterized by the number of ribosomes in the cell and $M_u$), size of cell load (characterized by $N_{cp}$), cell doubling (characterized by $t_{CD}$)) that affect the functioning of the entire cell. In that sense, these parameters can be considered as the design parameters for ab initio cell design—they allow the cell designer to specify the overall cell structure and behavior with a small number of fundamental parameters before diving into the task-specific smaller details.

It was shown based on the developed models that it is possible to estimate the value of $M_u$ which is one of the central parameters in CHD cell cycle theory. So far, the interpretation of the meaning of $M_u$ has been limited because it has been mostly handled as an empirical parameter of essentially

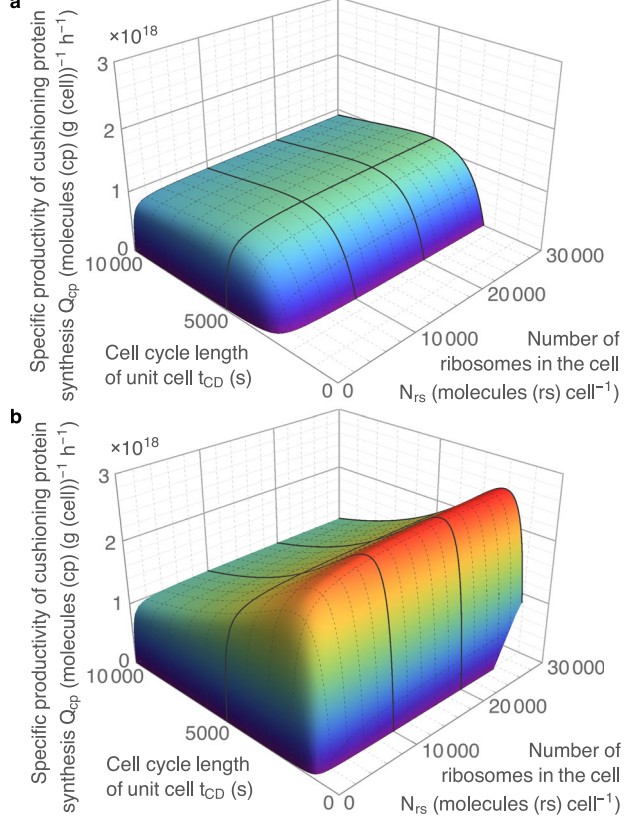

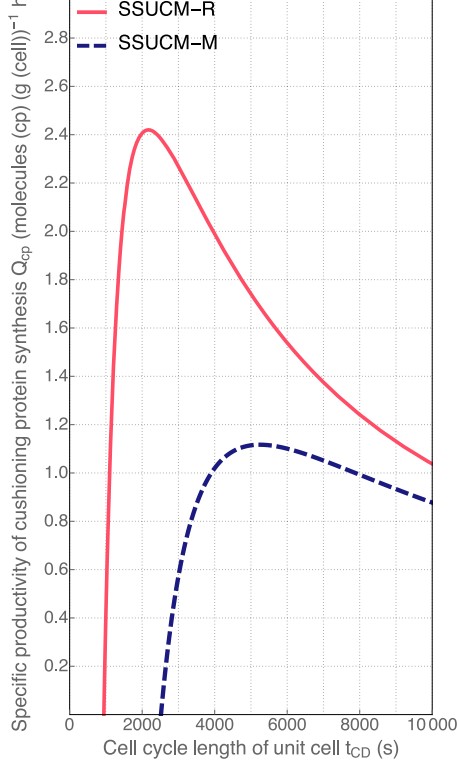

**Fig. 7 | Productivity of unit cells growing on minimal and rich media. a** The relationship between the specific productivity of cushioning protein synthesis ($Q_{cp}$, molecules (cp) (g (cell))$^{-1}$ h$^{-1}$), the number of ribosomes in the cell ($N_{rs}$, molecules (rs) cell$^{-1}$) and cell cycle length of the unit cell ($t_{CD}$, s). Calculations and regression surfaces were obtained for various $N_{rs}$ (<3·10$^4$ molecules (rs) cell$^{-1}$) and $t_{CD}$ values (<10$^4$ s = 2.8 h) using SSUCM-M. The visualized range of $N_{rs}$ corresponds to the

range of molar concentration of ribosomes $C_{rs}$ < 3.9·10$^{-5}$ mol (rs) L$^{-1}$. **b** Similar relationship in the case of SSUCM-R. The visualized range of $N_{rs}$ corresponds to the range of $C_{rs}$ < 7.1·10$^{-5}$ mol (rs) L$^{-1}$. **c** The relationship between $Q_{cp}$ and $t_{CD}$ in the cases of rich (solid line) and minimal medium (dashed line) at $N_{rs}$ = 10$^4$ molecules (rs) cell$^{-1}$ (equivalent to the range of $C_{rs}$ = 1.1·10$^{-5}$–3.7·10$^{-5}$ mol (rs) L$^{-1}$ for SSUCM-M and the range of $C_{rs}$ = 1.1·10$^{-5}$–7.0·10$^{-5}$ mol (rs) L$^{-1}$ for SSUCM-R).

arbitrary value. The current study provides the opportunity to estimate not only the value of $M_u$, but also all other main parameters of UC. This helps to put experimentally determined unit cell size values into the wider context in terms of other cellular processes, to ask questions about how $M_u$ itself is determined and to find possible growth boundaries of UCs. In the current work, the growth of larger UCs was limited by membrane surface area and the growth of smaller UCs was determined by the available space inside the cell.

It was shown that $M_u$ determines the essential features of the molecular physiology of UCs. According to the CHD theory, $M_u$ also determines the value of cell mass for the rest of the growth region where cell cycle length ≠ $t_{CD}$. So, it is possible to conclude that $M_u$ also determines the essential features of non-UCs, i.e., of basically all normally replicating bacterial cells (at least during steady-state growth).

But the relevant experimental data about $M_u$ is scarce and includes also relative units that can not be integrated into models without problems. The ideal experimental set-up must take into consideration that UCs are growing slowly. They grow in most cases certainly much slower than cells growing near the upper growth rate limits. It is certainly possible to transform the parameter values of fast-growing cells to values of UCs but it requires additional assumptions about calculation procedures and physiological parameter values (for example $M_u$ constancy for the growth range). Therefore, it is preferable to acquire the experimental data from actual slowly growing cells.

However, cultivation of slowly growing cells excludes methods operating near maximal growth rate (batch, turbidostat-family of continuous cultures) except certain defined minimal media consisting of carbon sources that do not enable fast growth. Apparently, the chemostat-family of continuous cultivation methods are much more suitable due to operating in substrate-limiting region.

The targeting of UC where DNA replication initiation takes place exactly at $t_0$ as in our models is obviously a very demanding task. One of the main problems is that $t_{CD}$ depends on the growth conditions which makes it difficult to select the right dilution rate. One possible approach could be to utilize the A-stat cultivation technique[56] with very low acceleration and starting from low initial dilution rate. Such experimental set-up ensures that the whole segment of growth range under study can be systematically scanned and the precise $t_{CD}$ found in combination with marker frequency analysis using qPCR.

However, there is an additional problem concerning the average cell concept of the asynchronous exponentially growing cell population. The targeting of $t_0$ is out of question considering the cell age distribution because the parameter values of an average cell are located in the middle of cell cycle. This problem can be solved by synchronization of the culture (effect of inhibitors) or by cultivation of single cells in microfluidic systems[57,58] (cell interactions and population heterogeneity are also avoided) but the stability of those systems is the main issue[59]. Alternatively, it might be possible to use live-cell imaging and marker technologies[60] to detect and monitor individual UCs in a small nonhomogeneous population.

As mentioned before (in section "Calculated unit cells based only on self-reproduction systems are problematic"), relatively slow growth is not the usual characteristic of SRS as it can be achieved only in the case of small numbers of cell component molecules/complexes in the cell. Therefore, it is obvious that a UC consisting of only SRS can exist also only in extreme cases. To enable the growth of UC without peculiarities, cushioning parameters must be introduced which allow to increase the size of the non-ribosomal part of the cell. One possible interpretation is that the cushioning parameter might be realized as (the amount of) molecules—for example proteins that are not directly necessary for self-replication. The results indicated that cushioning mass (or cellular parameters that affect the size of cells) was necessary to avoid too small cell sizes, unusual compositions and intracellular concentrations.

The identification of such cushioning parameters is important because separately they do not have a direct role for SRS but their main functions in terms of cell design principles are to slow down the SRS and to increase metabolic stability by increasing the numbers of cell components (if some important cell components are present only in very small numbers, then small fluctuations in their numbers can cause major metabolic disturbances). Based on that, it is possible to design faster chassis cells and to replace the default cushioning mass with biotechnologically useful components and products.

The described features of the cushioning parameter can be studied experimentally by deleting unnecessary (in terms of self-replication of the cell) genes from genomes. Similarly, biosynthesis pathways can be shortened or eliminated by genetic modifications leading to fewer cell components. Theoretically, minimization of complexity should lead to smaller cells, but it is not guaranteed if cells might have different mechanisms to maintain their cushioning mass. It must be noted that the deleting of genes and silencing of genes have a different theoretical effect on cell cycle because the former will probably decrease the value of $t_C$.

An approach for the selection of $t_{CDopt}$ for the production of CP was proposed for UC. In the case of $t_{CD}$ = const, $Q_{cp}$ of UC increased hyperbolically with the number of ribosomes in the cell, approaching asymptotically the maximal value. The dependence between the $Q_{cp}$ of UC and $t_{CD}$ resembles a bell-shaped Gauss distribution with a global maximum. The location and value of the $Q_{cp\_max}$ depended on the models (SSUCM-M, SSUCM-R) considered. In the case of a rich medium the $Q_{cp\_max}$ was higher and $t_{CDopt}$ lower because the complexity of the corresponding SRS was smaller than in the case of a minimal medium, thus leaving more room for CP. Our result that $Q_{cp\_max}$ can be achieved in UC at $t_{CD}$ that is approximately twice the minimal $t_{CD}$ is an important characteristic for cell designers to make more productive producer cells.

However, it must be emphasized that cushioning is an "invisible" function in the cell or even an abstract concept for explaining the distributed properties of the models—it is not directly linkable to the individual physical molecular characteristics of the components. Cell components that are not directly participating in the self-reproduction of the cell are hardly without visible functions (although metabolic databases are full of proteins with unknown or unconfirmed functions). On the contrary, they might have even more than one function, for example enzymes might catalyze different reactions and bind multiple substrates. In that sense, cushioning is metaphorically a common denominator to various categories of cell components including special secondary cellular functions (regulatory, structural, homeostasis, etc.), reserve materials and unused SRS parts (idle ribosomes, metabolic pathways and transporters) which allow to respond flexibly to various changing growth conditions.

Currently, it is not possible to state in the case of multiple cellular functions that cushioning would be the primary goal for an individual cell component. Probably it depends on how the different functions are correlated, but the amount of an individual cell component with a special function is more likely determined by that same special function. It is difficult to imagine that a regulatory protein whose molecule numbers determine further steps in the regulatory network might have an elevated level in the cell without an input signal (although the existence of non-functional regulatory proteins due to chemical modifications, mutations, unfolding or misassemblies can not be ruled out). But, no doubt, in certain growth conditions some of the cell components from the multitude belonging to cell load might be also optimized according to the cushioning function.

However, it also means that a primitive understanding of the mechanical selection and replacement of cell load components (for biotechnological purposes) probably leads to a shrinking growth range and unexpected metabolic changes due to those interrelated cellular functions. Moreover, part of the theoretical calculated cushioning mass is probably the effect of minimal and integer molecule levels, which means that this part can not be replaced at all (Supplementary Discussion 5.4). Also, the division of the functional cell into SRS and cell load revealed some ambiguities already for such simple models, as explained for example in Supplementary Discussion 5.5. Hardly would it be simpler or easier (rather impossible) for cells with large numbers of cell component species and heavily underdetermined metabolic networks.

It should be noticed that several issues mentioned in the Discussion of ref. 1 and below must be considered during the interpretation of the current work due to the limitations of the used models. The conclusions of the current work can only be extended to cells that have the cushioning parameter type defined by accumulating cell components in the cell. Therefore, these results do not include other diverse cases like cells producing substances secreted to the environment. However, it would not be difficult to build respective SSUCMs describing those other cases.

Also, the current models and results are only limited to UCs, which are basically single points in the complete growth range of the cells. The main focus of this study was the introduction of the concepts of UC and cushioning parameter. For that, only the dependencies of central cell parameters ($M_u$, number of ribosomes in the cell, $N_{cp}$, $Q_{cp}$, $t_{CD}$ (mostly by changing only $t_D$)) were characterized and analyzed. The integration of full cell cycle algorithms and a comprehensive analysis of cell parameters will be provided in future publications.

One particular concern is the validity of the CHD cell cycle theory itself indicated by several studies mentioned in Introduction. Three main cell size homeostasis strategies (adder, timer, sizer) have been proposed and CHD is classified under the sizer. Although sizer-like control has been observed in some single-cell experiments with *E. coli* (for example during slower growth[27] or in the case of certain carbon sources[61]), it certainly does not cover all experimental conditions and results. On the other hand, there are no cell cycle models or theories available today that would perform without problems with all the data collected up to now. If some other theory will emerge and gain consensus, then it is, in principle, possible to replace the CHD in our models.

For example, a double-adder model has been proposed[62] that views the cell cycle rather as a period between DNA replication initiations and where events are not triggered by any specific cell cycle length ($t_d$) values but are coordinated by two distinct molecular accumulators that keep track of cell length changes. DNA replication is initiated when critical cell length $\Delta L_1$ per origin has been synthesized and added since the previous replication initiation, and cell divides when a different, shorter critical cell length $\Delta L_2$ per origin has been synthesized and added since the previous DNA replication initiation. Cell length at DNA replication initiation is taken to be $L_i \cdot N_{ori}$, where $N_{ori}$ is the number of replication origins.

Our cell models are based on nonstochastic processes and steady-state growth conditions, which means that during the period from one division until the next division cell size gets exactly doubled. In such conditions the double-adder model can be simplified as well: $\Delta L_1 \cdot N_{ori}$ becomes equal also to the cell length synthesized during the classical cell cycle between two divisions and, furthermore, $\Delta L_1 \cdot N_{ori}$ becomes also equal to the cell length $L_0 \cdot N_{ori}$ at $t_0$. Also, $\Delta L_2 \cdot N_{ori}$ becomes equal to the length added during $t_C + t_D$ and it is basically a fraction $(t_C + t_D)/t_d$ of $L_0 \cdot N_{ori}$. At higher growth rates, the value of $N_{ori}$ becomes exponentially larger according to $2^i$ where i is an integer. Therefore, in such conditions $L_i \cdot N_{ori}$ can be viewed as an alternative for the initiation cell mass $M_i$, and $L_i$ is an alternative for $M_u$ (we might call it unit cell length). If the replication of a single genome starts exactly at $t_0$ as in our unit cell models, then $t_d = t_{CD} = t_C + t_D$ and $\Delta L_1 = \Delta L_2 = L_i$. And, naturally, cell length is directly related to cell mass through cell geometry formulas (Supplementary Eqs. (121)–(124) from ref. 1).

Therefore, under the assumptions of our models we can easily replace CHD in our models with the double-adder mechanism and there will be a direct correspondence to the results presented in this paper, just that instead of the unit mass there would be unit length—our models would allow to turn the otherwise empirical unit cell length into a parameter that is directly quantitatively related to all other cell parameters and would thus allow to use the unit cell length as one of the key cell design parameters. Certainly, if growth conditions are not stable (non-steady-state) and/or the intracellular processes are stochastic, then the situation is different: our cell models would need to be changed considerably and also the behavior of a cell with the double-adder mechanism might diverge considerably from the behavior of a cell with the CHD mechanism. But within the scope of application of our

current models, the unit cell concept holds for the double-adder mechanism as well.

In the context of cell cycle studies, the main advantage of our models is that they are not focused only on the cell cycle but are describing also other important cellular systems. Classical cell cycle studies rely mostly on cell dimensions (length, radius, size, area), growth rate and cell cycle periods, whereas the synthesis and doubling processes of cell components other than DNA are usually not included. If that classical small set of cell parameters has not succeeded in fully explaining cell size regulation, then it is not unimaginable that there are other responsible parameters or relations that are not explicitly determined in the usual cell cycle studies. Therefore, our models could be useful tools for expanding the scope of cell cycle studies. In the current paper, the main focus was on presenting the models as such tools rather than solving some specific cell cycle question.

In the wider context, however, the general motivation of this and previous work[1] is to construct a modeling framework and tools for bottom-up cell design and a playground for proto-cell synthetic biology that would help to better understand some of the fundamental relations in the cell (hence the carefully selected strong simplifications compared to those computational models that try to include all possible known details) and to test out various scenarios with different sets of cell components and different parameter values of those components. This means that the focus is not on finding best-fit empirical models to particular experimental datasets, but on providing a possibility to put together the basic components of a cell in various combinations and to explore the space of cell construction possibilities with the mindset of a cell designer. We believe that such a modeling approach would be highly relevant especially to the emerging scientific initiatives of building synthetic cells from scratch.

To summarize, the following main conclusions can be drawn from the analysis of the developed SSUCMs integrating the doubling of SRS and the UC concept from CHD cell cycle theory. The analysis revealed the main cell design parameters (complexity of SRS, size of SRS, size of cell load, cell doubling time) and their most important relations were illustrated. It was shown that it is possible to estimate the values of UC parameters including $M_u$ which is one of the central parameters in CHD cell cycle theory and determines essentially the size of all non-UCs. The deterministic estimation of $M_u$ enabled also to clarify possible fundamental factors determining growth boundaries (cellular space and cell membrane surface area). The analysis of cell load (CP) indicated that cushioning was necessary to get reasonable quantitative properties of UCs and to secure slow growth of large cells. Also, the highest productivity of CP synthesis was encountered generally in the cases of smaller SRS, larger $M_u$ and usually not at the fastest $t_{CD}$ values.

## Methods
### Description of models

The primary purpose of the current work was to improve the theoretical understanding of the UC concept in the context of cell design (as opposed to finding the smallest and best-fitting model for particular experimental datasets). To accomplish this, four different SSUCMs were used in the current work as the minimal set of models needed to cover the four main possible combinations of the complexity of SRS (small vs. large complexity) and the availability of cell load (CP is synthesized vs. not synthesized):

(1) SSUCM-SRS-M (Supplementary Discussion 5.10.1) describes the growth of UC that includes simplified metabolic and biosynthesis pathways characterizing relatively complex SRS. As a biological example, such UC corresponds to the growth on a minimal medium. It is expected that UC consists of only SRS components and CP is not synthesized in the cell.

(2) SSUCM-SRS-R (Supplementary Discussion 5.10.2) describes the growth of UC that does not include central metabolic and biosynthesis pathways, thus characterizing relatively small SRS. As a biological example, such UC corresponds to the growth on a rich medium. It is expected that UC consists of only SRS components and CP is not synthesized in the cell.

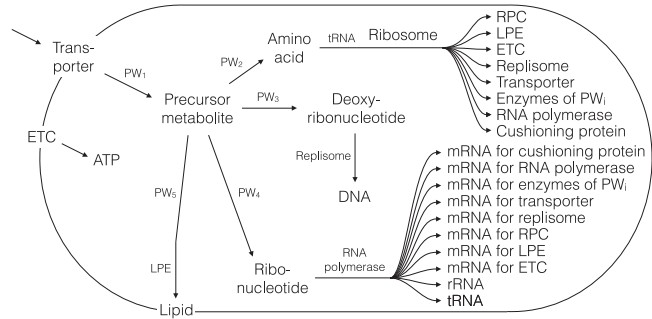

**Fig. 8 | Scheme of the bacterial unit cell (UC) with simplified metabolic network growing on minimal medium (the SSUCM-M model).** The developed SSUCM-M model is structured explicitly into cytoplasmic space and bilayer cell membrane of defined geometry. The model includes all the main cell components—macromolecules (DNA, RNA, proteins, membrane lipids) and their monomers (nucleotides, amino acids, lipids). The biopolymers are of the (labeled with terms *average, approximate, specific, precise*) length of corresponding *E. coli* polymers if possible (otherwise labeled with the term *generic*) and they consist of *average* (in terms of mass) monomers. An unspecified substrate is transported through the membrane by a membrane transporter and is converted to unspecified metabolic intermediates via a series of reactions of the central catabolic pathway PW₁. Energy is synthesized by the electron transport chain (ETC) complex on the membrane. The unspecified metabolic intermediates are used to synthesize monomers—amino acids, nucleotides, lipids—catalyzed by identical enzymes with generic amino acid sequence length through four different biosynthetic pathways PW₂–PW₅ consisting of sequentially arranged linear chains of reactions. The monomers and energy (ATP) are utilized by polymerization processes (replication, transcription, translation, membrane lipid synthesis). RPC is the ribosomal protein complex and LPE is the lipid synthesis enzyme. The stoichiometries of the energizing polymerization reactions are known from biochemistry. SSUCM-M includes also cushioning protein (CP) that is not directly necessary for the self-reproduction of the UC but fulfills a cushioning role. CP is accumulated in the cytoplasmic space and it represents unused SRS and cell load components.

(3) SSUCM-M (Fig. 8 and Supplementary Discussion 5.10.3) describes the growth of UC that includes simplified metabolic and biosynthesis pathways characterizing relatively complex SRS. As a biological example, such UC corresponds to the growth on a minimal medium. It is expected that UC includes cell load besides SRS components and CP is synthesized.

(4) SSUCM-R (Supplementary Fig. 58 and Supplementary Discussion 5.10.4) describes the growth of UC that does not include central metabolic and biosynthesis pathways. As a biological example, such UC corresponds to the growth on a rich medium. It is expected that UC includes cell load besides SRS components and CP is synthesized.

SSUCM-SRS-M and SSUCM-SRS-R were used to illustrate the problems and difficulties of growth of UCs with small cell size and extreme cellular composition. SSUCM-M and SSUCM-R were used to describe different concepts (cushioning parameter and load-bearing capacity) and to illustrate the effect of cell load (CP), SRS complexity and size on UC growth.

SSUCM-M is the most complex UC analyzed in this paper involving a set of cell components (DNA, RNA, proteins, membrane lipids) and their monomers (nucleotides, amino acids) visualized in Fig. 8. Most of the cell components are localized in the cytoplasmic space except membrane lipids, substrate transporter proteins and ETC complexes that belong to the bilayer cell membrane. The information about the localization of cell components is necessary for including cell geometry and size calculations in the models.

The genome of the cell is a bihelix circular chromosome. The RNA fraction consists of diverse RNA types (assembled rRNA subunits based on 5S, 16S and 23S sub-units with 1:1:1 stoichiometry; universal tRNA as different amino acids are not specified; mRNAs for all proteins containing

only coding regions). Different proteins are considered based on their functions (enzymes, polymerases, membrane proteins). Besides the proteins that are needed for self-reproduction, there are also some proteins (CP) that are not directly associated with self-reproduction. The latter concept is analogical to refs. 30,42,63,64 where such cell components have been named dummy, useless and unspecified proteins, respectively. It is expected that their involvement is necessary and covers distinct aspects of physiology (regulation, structure, signaling, housekeeping, unused metabolic proteins) that are not described explicitly in the models.

The developed SSUCMs are based on a number of main principles, simplifications and base assumptions. Some of them (steady state, pool sizes, parallel and continuous processes) have been described in Supplementary Discussions 5.11.1–5.11.2 of ref. 1. The remaining base assumptions of ref. 1 required slight modifications in the current work:

(1) As in ref. 1, $M_u$ and all cell components (numbers of cell component molecules/complexes in the cell) are exactly doubled during growth. Specifically, the UCs are doubled at the end of the cell cycle and the experimentally observed variability of cell division events is not included[65]. UC at $t_0$ is a mother UC and it is building a daughter UC during the cell cycle. As in ref. 1, this means that the numbers of cell components synthesized during the cell cycle are equal to the numbers of those same cell components existing in the cell at $t_0$. Therefore, $t_0$ is the most convenient time point for which to construct and solve the equation systems, and this is indeed how it is done in the used models. The numbers of cell components at all other cell age values can be calculated easily from their values at $t_0$, i.e. from the solutions of the equation systems of the models.

(2) As in ref. 1, $M_u$ and numbers of cell component molecules/complexes in the cell are growing during the cell cycle according to the linear growth law[31,66]. It is assumed that the numbers of active (mother) cell components (ribosomes, polymerases, enzymes, etc.) do not change during the cell cycle. Additional numbers of cell component molecules/complexes in the cell synthesized during the cell cycle are not active after the synthesis during the same cell cycle, they are activated only after the cell division in the daughter UC without specifying responsible mechanisms. It is also assumed that all processes follow the linear function, including periodic cell processes (DNA replication, cell division).

(3) As in ref. 1, optimized metabolism is assumed. At the beginning of the cell cycle ($t_0$), all cell components are active, they are participating in synthesis processes and there are no free cell components. Important exceptions to this assumption are periodic cell processes (such as DNA replication), and also CP and other related cell components that are not necessary for self-replication but fulfill a cushioning function in SSUCM-M and SSUCM-R (Supplementary Discussions 5.10.3–5.10.4).

(4) The description of a simplified cell cycle was introduced to the used models instead of simple biomass doubling. It is again expected that UCs include only a single genome and a pair of replisomes at $t_0$ but the context is different. In ref. 1 the growth of proto-cells (essentially the doubling of cellular components) was described, whereas the current work describes the growth of UCs. The concept of UC was derived and extended from the CHD cell cycle theory[2,3,13,14] and from the general growth law[28,37]. It is assumed that cell growth, division and DNA replication are coupled together in UC (section "Cell cycle of unit cells").

Due to these base assumptions, the developed modeling approach is stoichiometric, and therefore, all defined synthesis processes can be described by simple algebraic balances. The interactions between different cell components in the models reflect several necessary cellular processes and stoichiometric constraints. The mathematical descriptions of corresponding equations (Supplementary Eqs. (13)–(32), (39)–(48)), model parameters and their values (Supplementary Tables 12, 15, 16) are largely similar to those of simplified single-cell models of proto-cells[1].

The form of the balance equation for a general metabolic intermediate on a linear reaction chain (as in pathways $PW_1$–$PW_5$) is described by Eq. (3):

$$N_{cell\_comp\_cat} \cdot k_{cell\_comp} = N_{cell\_comp\_cat} \cdot k_{cell\_comp} \qquad (3)$$

where synthesis and degradation reactions are denoted by the number of catalyzing cell component molecules/complexes in the cell ($N_{cell\_comp\_cat}$) and their apparent working rate $k_{cell\_comp}$. Multiplications in Eq. (3) are equal to the synthesis and degradation fluxes of the general metabolic intermediate.

The polymerization (synthesis of proteins, RNA, DNA) rate depends on the apparent working rate of polymerase ($k_{pol}$) and $t_{CD}$. Considering the base assumptions, material balances for biopolymers can be written as a simple nonlinear equation (Eq. (4)):

$$t_{CD} = \frac{\sum N_{cell\_comp} \cdot n_{cell\_comp}}{N_{pol} \cdot k_{pol}} \qquad (4)$$

where $N_{cell\_comp}$ denotes the number of cell component molecules/complexes in the cell, $n_{cell\_comp}$ denotes the number of monomer molecules in the macromolecular cell component molecule/complex and $N_{pol}$ denotes the number of molecules of the corresponding polymerase (ribosome, RP complex and RC) in the cell. All biopolymers must be doubled during $t_{CD}$, and there must be enough catalysts. It must be stressed again that the total number of biopolymers (designated by $N_{cell\_comp}$) polymerized during $t_{CD}$ is precisely equal to the number of biopolymers at the beginning of cell cycle. Therefore, Eq. (4) describes the simplified doubling of biopolymers in UC.

Additional balances are related to the geometry and size of UCs. ETC on the cell membrane must produce the exact amount of ATP that is consumed by different monomer biosynthesis pathways ($PW_2$–$PW_5$), polymerization processes and substrate transport (Fig. 8) to meet the requirements of the precise doubling condition. The surface area of the cell (membrane) is the sum of surface areas of different membrane components (ETC, substrate transporter, membrane lipid) and it depends on the volume of cytoplasmic space of the cell.

It must be stressed that certain individual equations differ between individual models but the general equation forms are similar. The main differences are determined by the parameters of synthesis pathways, CP and enzymes. For example, SSUCM-M describes the polymerization of all mentioned proteins (symbols are explained in Supplementary Discussion 2):

$$t_{CD} = \frac{N_{pol} \cdot n_{pol} + N_{stp} \cdot n_{stp} + N_{etc} \cdot n_{etc} + n_{enz} \cdot \sum_{i=1}^{5} N_{enz\_PWi\_r} \cdot l_{PWi} + N_{cp} \cdot n_{cp}}{N_{rs} \cdot k_{rs}} \quad (5)$$

On the other hand, SSUCM-R does not include the synthesis of monomers of biopolymers (DNA, RNA, proteins) and, therefore, the corresponding polymerization equation does not include the translation of enzymes:

$$t_{CD} = \frac{N_{pol} \cdot n_{pol} + N_{stp} \cdot n_{stp} + N_{etc} \cdot n_{etc} + N_{cp} \cdot n_{cp}}{N_{rs} \cdot k_{rs}} \qquad (6)$$

SSUCM-SRS-M does include the polymerization of enzymes but not that of CP:

$$t_{CD} = \frac{N_{pol} \cdot n_{pol} + N_{stp} \cdot n_{stp} + N_{etc} \cdot n_{etc} + n_{enz} \cdot \sum_{i=1}^{5} N_{enz\_PWi\_r} \cdot l_{PWi}}{N_{rs} \cdot k_{rs}} \qquad (7)$$

Finally, SSUCM-SRS-R includes neither the polymerization of CP nor the enzymes:

$$t_{CD} = \frac{N_{pol} \cdot n_{pol} + N_{stp} \cdot n_{stp} + N_{etc} \cdot n_{etc}}{N_{rs} \cdot k_{rs}} \qquad (8)$$

Similar differences exist also for the equations describing synthesis pathways, energy consumption, cell mass balance, etc.

The parameters of equations are divided into input (predetermined) and output (calculated using the system of equations) parameters. The number of independent model cell parameters is approximately 1/3 larger than the number of independent equations in the case of SSUCM-M, which means that a smaller number of the parameters' values must be fixed (input model parameters) to get a determined system with a unique solution.

Input parameters for solving the models were chosen mostly among those cell parameters that have constant or relatively unchanging values for a given strain in a given environment (such as dimensions and compositions of molecules, stoichiometries of reactions and cellular processes, $k_{pol}$, etc.). The values of the input parameters of the used models are assumed in most cases to be equal to the characteristics of the respective molecules of *E. coli*. For example, the biopolymers (labeled with terms *average*, *approximate*, *specific*, *precise*) are of the length of corresponding *E. coli* polymers, if possible (otherwise labeled with term *generic*) and they consist of *average* (in terms of mass) monomers. The standard values of most of the input parameters are provided in ref. 1 and the remaining values in Supplementary Discussion 5.10. Most of the input parameters were considered constant during calculations except independent variables including $t_{CD}$ that was an independent variable for all used models with the range of $<10^4$ s = 2.8 h. The changes in $t_{CD}$ values reflected changes mainly in the values of $t_D$ and also in the apparent working rate of DNA polymerase (only in the case of faster growth of SSUCM-R) but not in the number of deoxyribonucleotide molecules in the genome (Supplementary Discussion 5.10.3.2.1). $N_{cp}$ was an independent variable for SSUCM-M and SSUCM-R models. The varied range was approximately $0$–$1.7 \cdot 10^7$ molecules (cp) cell$^{-1}$ and it corresponded to the $M_u$ range of $10^{-14}$–$10^{-12}$ g (unit cell)$^{-1}$ or to the range of molar concentration of cushioning protein of $0$–$0.028$ mol (cp) L$^{-1}$.

Output parameters of the models were chosen among those cell parameters that lack respective databases and have nonstatic nature (cellular parameters that depend on growth): numbers of cell component molecules/complexes in the cell, fluxes, cell compositions, geometric dimensions and sizes of cells. Sets of calculated cell parameter values have been presented for each model in Supplementary Tables 5–8, 13, 14.

The following calculations were carried out in the paper:

(1) Calculations based on constant standard values of input parameters in section "Calculated unit cells based only on self-reproduction systems are problematic" using models SSUCM-SRS-M and SSUCM-SRS-R.

(2) Calculations based on the variable $N_{cp}$ and constant standard values of other input parameters in section "Cushioning protein ensures physiologically reasonable cells" and Supplementary Discussion 5.7 using models SSUCM-M and SSUCM-R.

(3) Calculations based on the variables $N_{cp}$ and $t_{CD}$ and constant standard values of other input parameters in sections "Changing $t_{CD}$ has major effects on properties of unit cells" and "The productivity of cushioning protein synthesis in unit cells has optima" using models SSUCM-M and SSUCM-R.

(4) Calculations based on constant standard values of input parameters and experimentally determined values of $t_{CD}$ and cell size in Supplementary Discussions 5.2, 5.6 and 5.8 using models SSUCM-M and SSUCM-R. In addition, there are a few special cases such as calculations based on changed values of $k_{rs}$ and data of ref. 28 (mentioned in section "Changing $t_{CD}$ has major effects on properties of unit cells" and in Supplementary Discussion 5.8).

It must be stressed that SRSs are identical for models growing on the same type of medium, i.e. models within the pair (SSUCM-SRS-M, SSUCM-M) for minimal medium and models within the pair (SSUCM-SRS-R, SSUCM-R) for rich medium. It means that the solutions within those model pairs are identical at growth boundaries where $N_{cp} = 0$, for example in Fig. 2 at *approximate* $t_{CD} = 3520$ s = 1.0 h (Supplementary Discussions 5.10.3.3, 5.10.4.3).

This applies also when comparing the models in this paper to the proto-cell models SSPCM-M and SSPCM-R in ref. 1: the values of the minimal $t_{CD}$ of SSUCM-SRS-M are equal to the minimal doubling times of SSPCM-SRS-M, and the values of the minimal $t_{CD}$ of SSUCM-SRS-R are equal to the minimal doubling times of SSPCM-SRS-R.

Full descriptions of the used models are presented in Supplementary Discussion 5.10.

## Computations
The modeling procedures, tools of the used models and visualization tools of model schemes are described in Methods of ref. 1. Graphs were created with Mathematica 10 (Wolfram Research) and MATLAB (MathWorks) using several functions for the visualization of 3D surfaces and 2D graphs. Numerical source data for Figs. 1–7 is presented in Supplementary Data.

## Reporting summary
Further information on research design is available in the Nature Portfolio Reporting Summary linked to this article.

## Data availability
All data analyzed in this article are included in the article, its Supplementary Information and Supplementary Data.

## Code availability
The code of used models in Mathematica m-file format is provided in the Supplementary Software.

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

## Acknowledgements

This research was financially supported by the European Regional Development Fund project EU48667.

## Author contributions

K.Ab. and R.V. developed the theoretical basis for the used modeling framework. K.Ab., I.M. and R.V. constructed the models. K.Ab., N.A.E. and T.L. implemented the models. K.Ab., J.A. and P.Š. carried out model calculations and analyzed the results. N.A.E. and T.L. carried out model calculations, visualized the results and generated figures. All authors (K.Ab., K.Ad., J.A., N.A.E., T.L., I.M., A.S., P.Š., R.V.) participated in drafting the manuscript, contributed to the discussion and approved the final version before submission.

## Competing interests

The authors declare no competing interests.
