## [Transparent Peer Review file · Communications Biology]

The design of unit cells by combining the self-reproduction systems and metabolic cushioning loads

Corresponding Author: Professor Raivo Vilu

Version 0:

Reviewer comments:

Reviewer #1

(Remarks to the Author)

The article is based on previous research published by the authors about self-reproduction and doubling time limits of different cellular subsystems. The authors include the metabolic cushioning loads and aim to explore the effects of the number of cushioning protein molecules and number of ribosomes in the prediction of the unit cell mass used for different models of initiation of DNA replication.

I find that the article lacks self-contained explanations, and it frequently refers to supplementary information and previous works without providing clear motivation in simple language for non-specialized readers. The results are presented in comparison to those of the prior article. The authors explored four different models, and while the results differ only in terms of specific numerical values, they predict very similar trends. Given this similarity in trends, I did not discover new conceptual insights that help me intuitively grasp the relevance of these distinct models.

For instance, why should I study the SSUCM-R model and the SSUCM-M model separately if they both yield a similar pattern in Figure 2C? If the trends align closely, it seems more reasonable to favor the simplest model with slight rate modifications. Proposing complex models is only worthwhile if they significantly diverge from the predictions obtained using simpler models.

I suggest the authors to use simple schematic modes that help the reader to understand the connection between key parameters: Unit cell mass, cell age, cell cycle length, cell load, replication time. Also, what is the difference between the proposed models of unit cells with biological examples where each model can be applied.

With number of parameters used for the model I think it would be easy to fit observed data, for instance the initiation mass for cells in reference 28 for cells with different ribosome concentration seems to be very uniform. Does your model predict these datasets? What are the limitations in the prediction? The authors mention a debate on the existence of the initiation cell size. How the article contributes to this debate?

I think it is good to mention the design of experiments that help us to test the predictions of the model. Please use other articles for make the discussion richer. To me, the discussion currently focuses on the reference article and the studied plots.

The abstract did not show a strong conclusion with a simple explanation, please elaborate better the findings.

Discuss what modifications to the model has to be done if the reader wants to assume alternative models of replication initiation such as

Witz G, van Nimwegen E, Julou T. Initiation of chromosome replication controls both division and replication cycles in *E. coli* through a double-adder mechanism. *Elife*. 2019 Nov 11;8:e48063.

I also recommend working on decreasing the length of the article. It has multiple details, for instance the description of the

CHD model, the τ_c and τ_d that can be avoided using a good schematic. Also, the particular values of some parameters are not really relevant. I think from the perspective of modeling designing, it is relevant to highlight on how the variables interact and how these interactions differ model to model.

It is not clear the purpose of the model: Improve the theoretical understanding of the phenomena or making predictions. For the first case I would recommend simplifying the approach to get a minimal model with the least number of assumptions, present differential equations such as it is straight to get insights. Is the purpose being the second one, I would recommend fitting data or other studies.

Reviewer #2

(Remarks to the Author)

The authors extended the analysis done in Abner et. al (2023) and calculated several physiological parameters related to the Cooper-Helmstetter-Donachie cell cycle theory.

This is partly due to my lack of knowledge, but I have to say that the paper is unconstructive. I read through the paper and found several values and tons of abbreviations, but I cannot see what the real question is and what the take home message of the paper is.

I strongly recommend that the authors learn paragraph writing to reorganize the manuscript and resubmit it as a new submission. In my opinion, at least the main question, assumptions (or models), and results should be written very clearly. And I cannot believe that people can read papers with tons of abbreviations. It was a big surprise for me that more than 10 pages are dedicated to the description of abbreviations, symbols and definition of terms in SI text. Why do authors need so many symbols and abbreviations? For example, what is the point of abbreviating ribosomes as "rs"? It just confuses the readers, I would say.

Also, the authors are asking too much of the reader to refer to the SI text.

Overall, I see that some calculations have been done in the manuscript, but I cannot clearly see how these calculations have been done and what the values of the calculations are.

I am sorry that there are several points that I cannot follow up partly on due to my lack of knowledge, but I think that academic papers should be written with the authors' maximum effort to make the paper as readable as possible.

Version 1:

Reviewer comments:

Reviewer #1

(Remarks to the Author)

The resubmitted version of the article shows a notable improvement in most of the concerns.

The article is easier to follow, and each model is well justified. Although the article is relatively long, I think it explains most of the needed information. I just have a couple of recommendations:

Would you discuss typical parameters for the observed parameters in the main article as a reference to compare with the results. What is approximately value of the initiation mass in a bacterium (or cell volume assuming a typical cell density) and the typical number of ribosomes in a bacterium. The authors say 'slow' or 'small' but it is not clear you reference.

I recommend using the same units of time and less significative digits. I would use the cell cycle time in hours if you were in the slow-growth regime. Experimentally does not make sense to have an accuracy of decimal of second.

I think it is needed to define a couple of concepts that are not clear specially to the community of mathematical modeling: Proto-cell, the role and mechanisms of ribosomes, cell load, cell cushioning.

As an optional change, I think it is possible to approach the analysis instead using number of molecules, molecule concentration. This to have a better understanding using traditional units.

I would finally recommend to add a conclusions paragraph at the end of the discussion to highlight quickly the main findings.

Reviewer #2

(Remarks to the Author)

The authors have made improvements in the manuscript's writing compared to the previous version. However, the text remains quite complex and does not adhere consistently to the principles of academic writing, such as dedicating each paragraph to a single topic. As a result, the manuscript is still difficult to read.

Nonetheless, the revisions and responses have clarified the manuscript's focus for me. My major concerns are as follows:

1. The model description should be clearly presented within the main manuscript rather than relegated to the Supplementary Information (SI). For instance, in line 297, the authors state: "First, we calculated the μ and other parameters of UC at approximate $t_{CD} = 3519.84 \pm 296$ using SSUCM-SRS-M and SSUCM-SRS-R, which are just SRSs at unit cell condition." This explanation is too brief and lacks sufficient detail.
2. The stated objectives of the manuscript—to "expand the analysis," "provide an opportunity to link..." and "give the theoretical basis" (lines 109-119)—are vague and do not constitute clear scientific questions. It is always possible to argue the expansion of any analysis or the provision of any opportunity, for instance just by adding one more data point. The authors clarify what question is revealed by this expansion.
3. The authors claim that constructing a minimal model is not their goal. However, the model presented is neither detailed (like a whole-cell model) nor minimal. If a model is detailed and realistic, its parameters should be empirically derived. Conversely, if a model is minimal and coarse-grained, parameter selection can be somewhat arbitrary, though the model's flexibility is limited by the small number of variables. Both approaches offer valuable insights. However, an intermediate model, which sits between realistic and minimal, risks reproducing any result due to the lack of constraints on the number of variables and parameter choices. I find it challenging to see the value in the outcomes from such a model.
4. This point relates to my 3rd comment. The introduction of cushioning parameters appears to be a way to align the calculation results with existing literature. While I acknowledge that abstract proteins, like cushioning proteins, are sometimes introduced in models, particularly in proteome-allocation physiology research, these models are usually minimal. For example, in Scott et al., *Science* (2010), the introduction of class-Q proteins is not crucial to the main argument.
5. The results presented seem mathematically trivial. The authors should emphasize the biological novelty of their findings, perhaps by demonstrating that the model fits well with a substantial portion of the data. However, this is not adequately presented in the manuscript (I checked SI. But I cannot see the figures supporting the model "prediction". The figure just shows the calculated values by the model, rather than comparing the model prediction and data?).
6. Overall, the authors explore the model's dependency on the parameters, but I cannot see why we need to know it, because the reliability of the model to understand biology is not presented.

Regarding the authors' rebuttal letter: They respond by stating that "rs" is a common abbreviation in the field and that "Genome-scale models are more complicated than ours." I encourage the authors to be more constructive in their responses. My point is that readers are required to remember numerous abbreviations. Considering the cognitive load on readers, what justifies the use of the abbreviation "rs"? My intention in discussing the number of abbreviations was to highlight that the manuscript is difficult to read. Readability and reproducibility are not mutually exclusive. Many well-written genome-scale studies effectively present their questions, methods, results, and conclusions without relying on an excessive number of abbreviations. Detailed information can be provided separately for those interested in reproducing the results. Arguing that "this approach is acceptable because another paper has done it" is not a constructive rebuttal.

Version 2:

Reviewer comments:

Reviewer #1

(Remarks to the Author)

Most of my concerns were addressed.

The remaining concern is related to the time units. The authors discussed that the relevant time scale was seconds. However, Figure 6 and 7 have units of hours.

I would recommend some other minor style changes.

Use subsections with title being a summary of the main finding using a phrase. This helps to follow better the article and motivate the reader to follow the reading.

Given the length of the article, it is good to use a concluding sentence at the end of each section and a motivation to the next one such as the reader have clear why you structured the article the way you presented and what are the take-home messages. It will not be possible to read the article quickly, therefore you need to provide the reader pauses along the text.

Reviewer #2

(Remarks to the Author)

The authors have provided sufficient responses and revisions to clarify the issues and improve the overall quality of the manuscript. I find the current version satisfactory and recommend it for acceptance.

Responses to reviewers' concerns

Reviewers' comments are written in Times New Roman and answers to reviewers are written in italicized Calibri.

Reviewer #1 (Remarks to the Author):

The article is based on previous research published by the authors about self-reproduction and doubling time limits of different cellular subsystems. The authors include the metabolic cushioning loads and aim to explore the effects of the number of cushioning protein molecules and number of ribosomes in the prediction of the unit cell mass used for different models of initiation of DNA replication.

1. I find that the article lacks self-contained explanations, and it frequently refers to supplementary information and previous works without providing clear motivation in simple language for non-specialized readers. The results are presented in comparison to those of the prior article.

Indeed, the submitted article was organized so that a substantial part of the content and explanations were available in the previous article or in the Supplementary Information considering the volume of this work. But it seems that we relied on this approach too much and we did not take into account possible difficulties for the readers. Based on that, we have now reorganized the whole text of the article and replaced the referring sentences by longer and simpler explanations as much as possible.

Concerning the comparison of the results from previous (Abner et al., 2023) and current work, references to the previous article were included mainly due to following reasons:

- a. If specific original terms (like self-reproduction system (SRS)) were thoroughly explained in the previous article and it was necessary to give sufficient background in the current article.*
- b. If some parameter values, equations, model details from the previous article were also used in the current work.*
- c. If some of the calculation results in the current article were determined by the properties of SRSs (such as the dependence between the doubling time and complexity of SRS) which have been thoroughly explained in the previous article.*

It seems that such a presentation of the article content emphasized too much the relationship between previous and current work. The actual overlapping part of both works is only the self-reproduction system. Based on that, we moved most of the technical references to the section "Description of models" and the SRS references to the section "Cell cycle of unit cells" in the resubmitted article.

2. The authors explored four different models, and while the results differ only in terms of specific numerical values, they predict very similar trends. Given this similarity in trends, I did not discover new conceptual insights that help me

intuitively grasp the relevance of these distinct models.

Indeed, the dependencies between calculated unit cell parameters (like unit cell mass or the number of ribosomes in the cell) and variables (like the cell cycle length of unit cell or the number of cushioning proteins in the cell) had in general quite similar trends. However, our main aim was to map quantitative differences between trends of the extreme biological cases described by the models. Those extreme biological cases can be interpreted as the values of cell design parameters. Therefore, quantitative differences between models show the dependencies between unit cell parameters and cell design parameters.

In this work, those cell design parameters are the complexity of SRS (the number of species of cell components that are required for the self-replication of SRS) and cell load (cell components that are not required directly for cell self-replication). The complexity of SRS can be interpreted in the used models as the length of metabolic pathways (the number of reactions per pathway) and growth on different media (minimal and complex), whereas cell load is described by the synthesis of cushioning protein. The calculations showed that lower complexity of SRS was associated with faster growth, smaller size of SRS and bigger potential for cell load synthesis. The synthesis of cushioning proteins enabled bigger cells and SRS size (the number of respective cell components like ribosomes) especially during slower growth. These trends between models were actually one of the main results from this work for us but it seems that these ideas were somehow lost in the text.

Based on that, we added to the end of the Introduction of the resubmitted article a short list of the main results, including also the SRS complexity and cell load issues. In addition, we added a section to the beginning of Results and Discussion trying to clarify the importance of those issues and the respective models. The same idea is included also in the section "Description of models" (p 38-39, lines 808-813).

3. For instance, why should I study the SSUCM-R model and the SSUCM-M model separately if they both yield a similar pattern in Figure 2C? If the trends align closely, it seems more reasonable to favor the simplest model with slight rate modifications. Proposing complex models is only worthwhile if they significantly diverge from the predictions obtained using simpler models.

Figure 3c depicts actually a very peculiar issue between SSUCM-R and SSUCM-M models and it is related to the selected cell load example in those models. The dependencies between calculated unit cell mass (M_u) and cell cycle length of unit cell (t_{CD}) are nearly identical at constant number of ribosomes in the cell (N_{rs}) although the models have clearly different SRS complexity (SSUCM-R does not include metabolic pathways and their enzymes). Because SSUCM-R must have smaller SRS size, it has larger potential for cushioning protein synthesis which is described in Figures 4c and 5c. However, the ratio of N_{rs} to M_u is almost identical between the models because, roughly, those ribosomes that are synthesizing enzymes in SSUCM-M must synthesize cushioning protein in SSUCM-R. Evidently, the mass of synthesized enzymes in SSUCM-M is equal to the mass of synthesized cushioning protein in SSUCM-R. The same N_{rs} value means that the number of RNAs, RNA polymerases and substrate transporters is also almost the same between the models. However, SSUCM-R does not need so much energy compared to SSUCM due to biosynthesis

costs. Therefore, the membrane compositions are very different between the models resulting in the 1 % difference between corresponding M_u values. In conclusion, there are clear quantitative similarities between the models (like N_{rs}/M_u and total protein content) but also clear differences (the number of lipids, energy synthesis, protein fractional composition, the number of cushioning proteins) if cell load is intracellular protein. Therefore, the trends align closely only for certain parameters. However, if cell load would have been defined as glycogen, then N_{rs}/M_u would have been also different between the models.

It seems that we failed to explain the meaning of Figure 3c properly and additional explanations are now included in the text of the resubmitted article (p 24-25, lines 478-490).

Calculated unit cell parameter values for SSUCM-M and SSUCM-R at $t_{CD} = 3519.84$ s. Largest differences in bold.

	SSUCM-M	SSUCM-R
Nrrna	10000	10000
Nrp	359.03	358.746
Ntrna	50000	50000
Nlpe	16.9036	23.5346
Nlip	5.95E+06	8.28E+06
RP/RS	0.00284279	0.00284054
	7.20358E-	1.00294E-
LPE/RS	06	05
Ncp	721409	1756390
Netc	14622.2	9814.03
Nstp	2186.88	2193.4
ETC/RS	0.207711	0.13941
STP/RS	0.00621301	0.00623153
Mcyt	4.70E-13	5.01E-13
CP/RS	0.307433	0.748495
Mu	5.82E-13	5.87E-13

As already mentioned in the second point, the models were not selected based on the divergent trends between cellular parameters but those models themselves were describing different combinations of SRS complexity and cell load values. We have used exclusively only simplified models in the current work but the general aim is to explain and develop a theoretical basis for larger models (strain-specific genome-scale models) which are useful in cell design and bioprocess optimization experiments. Naturally, the number of reactions or length of pathways is an important property in larger models. Therefore, we included such aspects also in the present simplified models. In our opinion, simpler models are useful for clearer presentation and explanation of the fundamental relationships, whereas complex models have practical importance for specific cells.

4. I suggest the authors to use simple schematic modes that help the reader to understand the connection between key parameters: Unit cell mass, cell age, cell cycle length, cell load, replication time.

We created an additional Supplementary Figure 1 that describes simple relations between the key parameters: M_w , N_{rs} , the number of cushioning proteins in the cell (N_{cp}), t_{CD} . Also, we included additional Supplementary Discussion 5.1 describing those relations in the Supplementary Information. Cell age is not analyzed in the work and the calculation results of the constructed models are always connected to the cell age at the beginning of cell cycle (t_0). Genome replication period is constant for almost all calculations except for faster growth on rich media. Varying of t_{CD} is carried out mainly by varying the value of t_D .

5. Also, what is the difference between the proposed models of unit cells with biological examples where each model can be applied.

As already mentioned in the second point, models were not selected based on the divergent trends between cellular parameters but those models themselves were describing the SRS complexity and cell load values. The selected minimal set of models that covers the main possible combinations of the complexity of SRS (small vs. large complexity characterizing rich vs. minimal media growth) and the availability of cell load (CP is synthesized vs. not synthesized):

- a. SSUCM-SRS-M has large SRS complexity and no cell load. This model describes the growth of unit cell on minimal media and the cell includes biosynthesis pathways for monomers of biopolymers. Cushioning protein is not synthesized and only essential genes for self-replication are active.*
- b. SSUCM-SRS-R has small SRS complexity and no cell load. This model describes the growth of unit cell on complex media and the monomers of biopolymers are transported from the growth media. Cushioning protein is not synthesized and only essential genes for self-replication are active.*
- c. SSUCM-M has large SRS complexity and cell load is included. This model describes the growth of unit cell again on minimal media and the cell includes biosynthesis pathways for monomers of biopolymers. Cushioning protein is synthesized and accumulated to the cell.*
- d. SSUCM-R has small SRS complexity and cell load is included. This model describes the growth of unit cell again on complex media and the monomers of biopolymers are transported from the growth media. Cushioning protein is synthesized and accumulated to the cell.*

There are two different SRSs which means that SRSs between model pairs are identical (SSUCM-SRS-M and SSUCM-M, SSUCM-SRS-R and SSUCM-R) and identical cushioning protein is synthesized in SSUCM-M and SSUCM-R.

We rewrote the corresponding text in the section "Description of models" (p 39, lines 814-833) and outlined the differences between the models more thoroughly in the resubmitted article.

6. With number of parameters used for the model I think it would be easy to fit observed data, for instance the initiation mass for cells in reference 28 for cells

with different ribosome concentration seems to be very uniform. Does your model predict these datasets? What are the limitations in the prediction?

The general motivation of this and previous work came from the principles of bottom-up cell design — how the quantitative properties of individual cell components and processes affect the growth and properties of cells. The primary purpose of the current work was to improve the theoretical understanding of the UC concept in the context of cell design by integrating together SRS and cell cycle (as opposed to finding the smallest and best-fitting model for particular experimental datasets).

Nevertheless, we now used the data (cell size, growth rate, cell cycle periods) from the literature (including (Si et al., 2017)) to demonstrate the capabilities of the models in the resubmitted article. We calculated the unit cell parameter values using SSUCM-M and SSUCM-R and literature data. The most important parameter values (M_w , cushioning mass (M_{cp}), RNA/protein (M_{rna}/M_{prot}), N_{rs} , etc.) together with tables and figures are included in the Supplementary Information (Supplementary Discussions 5.2, 5.6, 5.8), and shorter summaries of these calculations are added to the main text of the article (in section “The effects of cushioning protein” (p 19, lines 401-414), in section “The effects of changing t_{CD} ” (p 26-27, lines 522-553)).

There was a good match between the general trends of our theoretical results and the calculations based on experimental data. The comparison of calculated and measured values of the ratio of M_{rna}/M_{prot} showed that the general trend of the dependence between M_{rna}/M_{prot} and growth rate was similar (Supplementary Figure 8). However, the numerical values of the calculated M_{rna}/M_{prot} in the case of standard input parameter values were considerably lower than experimentally determined values. The reasons for such discrepancy might form a long list — the inability of simplified models to describe precisely the specific strains (metabolic networks, missing cell wall components), the assumption of cushioning mass being composed of only CP and excluding other possibilities like inactive ribosomes, growth-dependent cell parameter values, etc.

It must be noted that the models in the current work describe only unit cells ($t_d = t_{CD}$, only single genome at t_0), whereas the data from the literature were collected from cells growing in batch or turbidostat cultures which means that usually $t_d < t_{CD}$ and overlapping genome replications happened. Therefore, it was not reasonable to fit the M_{rna}/M_{prot} from (Si et al., 2017) except for few, almost overlapping (relatively close t_d and t_{CD} values), data points (MG1655 strain growing on MOPS glycerol and on MOPS glycerol supplemented with 0.2 mM uracil). Further calculations with those data points showed (Supplementary Figure 9) that almost perfect fit was possible for M_{rna}/M_{prot} if, for example, the value of k_{rs} was lowered from 20 to 4 – 5 molecules (aa) $s^{-1} rs^{-1}$. Most probably, the discrepancy is due to a combination of various factors and strain-specific larger models are needed for clarification. Constructing such larger models is indeed also possible with our modelling framework, but that was not the aim of this paper.

7. The authors mention a debate on the existence of the initiation cell size. How the article contributes to this debate?

The debate was reviewed to demonstrate that we are aware of open questions concerning cell cycle and that CHD theory has lost ground. However, the aim was not to specifically study bacterial cell size regulation but rather to propose new tools. Classical cell cycle studies rely mostly on cell dimensions (length, radius, size, area), growth rate and cell cycle periods, whereas the synthesis and doubling processes of cell components other than DNA are usually not included. If that classical small set of cell parameters has not succeeded in fully explaining cell size regulation, then it is not unimaginable that there are other responsible parameters or relations that are not explicitly determined in the usual cell cycle studies. We believe that our models might help to expand the scope in the field of cell cycle studies (but, as already mentioned earlier, the main motivation for constructing these models is not to focus on the specific details of cell cycle but to build a general cell modelling framework for cell design).

The described ideas are now included in the Discussion (p 37-38, lines 784-793).

8. I think it is good to mention the design of experiments that help us to test the predictions of the model.

There are several issues that must be considered in the framework of the developed models to design possible experiments for the study of unit cells. Firstly, unit cells are growing usually slowly. They are in most cases certainly much slower than cells growing near the upper growth rate limits. It is certainly possible to transform the parameter values of fast-growing cells to values of UCs but it requires an additional layer of calculation procedures and a series of assumptions (for example M_u constancy for the growth range). Therefore, it is preferable to acquire the experimental data from actual slowly growing cells. However, cultivation of slowly growing cells excludes methods operating near maximal growth rate (batch, turbidostat-family of continuous cultures) except certain defined minimal media consisting of carbon sources that do not enable fast growth. Apparently, the chemostat-family of continuous cultivation methods are much more suitable due to operating in substrate-limiting region. The targeting of UC where DNA replication initiation takes place exactly at t_0 as in our models is obviously a very demanding task. One of the main problems is that t_{CD} depends on the growth conditions which makes it difficult to select the right dilution rate. One possible approach could be to utilize the A-stat cultivation technique with very low acceleration and starting from low initial dilution rate (Adamberg et al., 2015). Such experimental set-up ensures that the whole segment of growth range under study can be systematically scanned and the precise t_{CD} found in combination with marker frequency analysis using qPCR. However, there is an additional problem concerning the average cell concept of the asynchronous exponentially growing cell population. The targeting of t_0 is out of question considering the cell age distribution because the parameter values of an average cell are located in the middle of cell cycle. This problem can be solved by synchronization of the culture (effect of inhibitors) or by cultivation of single cells in microfluidic systems (cell interactions and population heterogeneity are also avoided) (Jakiela et al., 2013; Long et al., 2013) but the stability of those systems is the main issue (Schmitz et al., 2019). Alternatively, it might be possible to use live-cell imaging

and marker technologies (Dovrat et al., 2018) to detect and monitor individual UCs in a small nonhomogeneous population.

The described features of the cushioning parameter can be studied experimentally by deleting unnecessary (in terms of self-replication of the cell) genes from genomes. Similarly, biosynthesis pathways can be shortened or eliminated by genetic modifications leading to fewer cell components. Theoretically, minimization of complexity should lead to smaller cells, but it is not guaranteed if cells might have different mechanisms to maintain their cushioning mass. It must be noted that the deleting of genes and silencing of genes have a different theoretical effect on cell cycle because the former will probably decrease the value of t_c .

The described ideas are now included in the Discussion (p 32-33, lines 641-667; p 33-34, lines 684-691).

9. Please use other articles for make the discussion richer. To me, the discussion currently focuses on the reference article and the studied plots.

Previously, part of the discussion of the submitted article consisting of references to other articles was stored in Supplementary Information (SI) and was hidden behind SI references (causing annoyance as mentioned in the first point). Considering this, we now replaced the corresponding sentences that referred to SI with sections that explicitly include at least part of the literature references directly in the main text (section "Calculated unit cells based on self-reproduction systems"). In addition, the changes described above in points 6 and 8 also substantially increased the number of article references in other sections of the main text.

10. The abstract did not show a strong conclusion with a simple explanation, please elaborate better the findings.

We now rewrote the Abstract to emphasize better the main results of the paper.

11. Discuss what modifications to the model has to be done if the reader wants to assume alternative models of replication initiation such as

Witz G, van Nimwegen E, Julou T. Initiation of chromosome replication controls both division and replication cycles in *E. coli* through a double-adder mechanism. *Elife*. 2019 Nov 11;8:e48063.

In the double-adder model, cell cycle is viewed as a period between DNA replication initiations and events are not triggered by any specific cell cycle length (t_d) values but are coordinated by two distinct molecular accumulators that keep track of cell length changes. DNA replication is initiated when critical cell length ΔL_1 per origin has been synthesized and added since the previous replication initiation, and cell divides when a different, shorter critical cell length ΔL_2 per origin has been synthesized and added since the previous DNA replication initiation. Cell length at DNA replication initiation is taken to be $L_i \cdot N_{ori}$, where N_{ori} is the number of replication origins.

Our cell models are based on nonstochastic processes and steady-state growth conditions, which means that during the period from one division until the next division cell size gets exactly doubled. In such conditions the double-adder model can

be simplified as well: $\Delta L_1 \cdot N_{ori}$ becomes equal also to the cell length synthesized during the classical cell cycle between two divisions and, furthermore, $\Delta L_1 \cdot N_{ori}$ becomes also equal to the cell length $L_0 \cdot N_{ori}$ at t_0 . Also, $\Delta L_2 \cdot N_{ori}$ becomes equal to the length added during $t_c + t_D$ and it is basically a fraction $(t_c + t_D)/t_d$ of $L_0 \cdot N_{ori}$. At higher growth rates, the value of N_{ori} becomes exponentially larger according to 2^i where i is an integer. Therefore, in such conditions $L_i \cdot N_{ori}$ can be viewed as an alternative for the initiation cell mass M_i , and L_i is an alternative for M_u (we might call it unit cell length). If the replication of a single genome starts exactly at t_0 as in our unit cell models, then $t_d = t_{CD} = t_c + t_D$ and $\Delta L_1 = \Delta L_2 = L_i$. And, naturally, cell length is directly related to cell mass through cell geometry formulas (Supplementary Eqs. (121)-(124) from (Abner et al., 2023)). Therefore, under the assumptions of our models we can easily replace CHD in our models with the double-adder mechanism and there will be a direct correspondence to the results presented in this paper, just that instead of the unit mass there would be unit length — our models would allow to turn the otherwise empirical unit cell length into a parameter that is directly quantitatively related to all other cell parameters and would thus allow to use the unit cell length as one of the key cell design parameters. Certainly, if growth conditions are not stable (non-steady-state) and / or the intracellular processes are stochastic, then the situation is different: our cell models would need to be changed considerably and also the behaviour of a cell with the double-adder mechanism might diverge considerably from the behaviour of a cell with the CHD mechanism. But within the scope of application of our current models, the unit cell concept holds for the double-adder mechanism as well.

In this simplified illustration, $N_{ori} = 1$.

We now added this discussion about the assumptions and changes necessary for the integration of (Witz et al., 2019) double-adder mechanism to our models to the Discussion (p 36-37, lines 753-783).

12. I also recommend working on decreasing the length of the article. It has multiple details, for instance the description of the CHD model, the t_c and t_d that can be avoided using a good schematic. Also, the particular values of some parameters are not really relevant.

The section "Cell cycle of unit cells" is now completely rewritten and sections describing the CHD model and its parameters are replaced by an additional Figure 1. Figure 1 includes a graph depicting the synthesis of the genome and doubling of cell size in a slowly growing cell during the cell cycle. Also, DNA replication initiation moments at different cell size and growth rate values are visualized. We removed the ranges of M_u values in the same section from the text.

Considering article length, the changes connected to the previous comments (more detailed model descriptions, improvement of readability, inclusion of literature) actually required longer texts. However, at least part of the added text we put into the Methods section which has unlimited space in the journal to ensure reproducibility (<https://www.nature.com/documents/CommsBio-file-checklist-revision.pdf>). To decrease the length of the article, more of the content should be relocated to Supplementary Information but that would also create problems for reviewers and readers.

13. I think from the perspective of modeling designing, it is relevant to highlight on how the variables interact and how these interactions differ model to model.

As already mentioned under the points 4 and 12, we included additional Supplementary Discussion 5.1 and rewrote the section "Cell cycle of unit cells" to explain more concisely the relations between the key parameters (t_{CD} , N_{rs} , N_{cp} , M_u , other cell cycle parameters). We expanded the section "Description of models" and wrote an additional section in Results and Discussion in order to clarify the reasons behind different models as already explained under the points 2 and 5. In addition, we included and explained examples of balance differences between the models in the section "Description of models" (p 45, lines 947-961). For example, the respective translation balances by models are the following:

SSUCM-SRS-R does not include monomer biosynthesis and cushioning protein synthesis:

$$t_{CD} = \frac{N_{pol} \cdot n_{pol} + N_{stp} \cdot n_{stp} + N_{etc} \cdot n_{etc}}{N_{rs} \cdot k_{rs}}$$

SSUCM-SRS-M includes monomer biosynthesis but not cushioning protein synthesis:

$$t_{CD} = \frac{N_{pol} \cdot n_{pol} + N_{stp} \cdot n_{stp} + N_{etc} \cdot n_{etc} + n_{enz} \cdot \sum_{i=1}^5 N_{enz_PW_i_r} \cdot l_{PW_i}}{N_{rs} \cdot k_{rs}}$$

SSUCM-R does not include monomer biosynthesis but includes cushioning protein synthesis:

$$t_{CD} = \frac{N_{pol} \cdot n_{pol} + N_{stp} \cdot n_{stp} + N_{etc} \cdot n_{etc} + N_{cp} \cdot n_{cp}}{N_{rs} \cdot k_{rs}}$$

SSUCM-M includes monomer biosynthesis and cushioning protein synthesis:

$$t_{CD} = \frac{N_{pol} \cdot n_{pol} + N_{stp} \cdot n_{stp} + N_{etc} \cdot n_{etc} + n_{enz} \cdot \sum_{i=1}^5 N_{enz_PW_i_r} \cdot l_{PW_i} + N_{cp} \cdot n_{cp}}{N_{rs} \cdot k_{rs}}$$

Similarly, there are differences in other balances.

However, the interactions between parameters and differences between the models are thoroughly presented in the code of the models (Supplementary File) and in Supplementary Discussion 5.10 (description of models).

14. It is not clear the purpose of the model: Improve the theoretical understanding of the phenomena or making predictions. For the first case I would recommend simplifying the approach to get a minimal model with the least number of assumptions, present differential equations such as it is straight to get insights. Is the purpose being the second one, I would recommend fitting data or other studies.

The general motivation of this and previous work came from the principles of bottom-up cell design — how the quantitative properties of individual cell components and processes affect the growth and properties of cells.

The main purpose of the previous work was to improve the theoretical understanding of design principles concerning SRS (including theoretical doubling time limits and the role of ribosomes), whereas in this paper the objective is a more complete cell (SRS integrated with cell cycle) in the framework of the unit cell concept. We still believe that it is not reasonable to replace the used combination of models with a minimal model because our aim was also to study the design parameters of cells (complexity of SRS and cell load) as already explained under the point 2.

Differential equations are good for dynamic models that explore the progression of cellular processes in time, which is indeed one very valuable perspective for understanding the cells. But another valuable perspective is to rapidly explore the possible diversity of cells with different components and different properties of those components (i.e. the cell design space), and our models allow to do that in a computationally very efficient way. The downside of that is that these models only provide a steady-state solution instead of dynamic processes, but the upside is that it is possible to get the full details (i.e. the values of all variables included in the model: the numbers of all molecules, sizes of fluxes, etc.) of any such steady state from anywhere in the cell design space almost instantly, which allows to conduct large-scale automated scans of the cell design space. A widespread example of a technically somewhat similar approach is Flux Balance Analysis (FBA).

We now added calculations based on different published datasets to illustrate the general modelling principles and capabilities of these models as already described under the point 6. However, our main focus is not on finding best-fit models to particular experimental datasets, but on constructing a modelling framework for bottom-up cell design that would help to better understand some of the fundamental relations in the cell and to test out various scenarios with different sets of cell

components and different parameter values of those components, i.e. to explore the space of cell construction possibilities with the mindset of a cell designer. We believe that such a modelling approach will be highly relevant especially to the currently emerging scientific initiatives of building synthetic cells from scratch.

Reviewer #2 (Remarks to the Author):

The authors extended the analysis done in Abner et. al (2023) and calculated several physiological parameters related to the Cooper-Helmstetter-Donachie cell cycle theory.

1. This is partly due to my lack of knowledge, but I have to say that the paper is unconstructive. I read through the paper and found several values and tons of abbreviations, but I cannot see what the real question is and what the take home message of the paper is.

I strongly recommend that the authors learn paragraph writing to reorganize the manuscript and resubmit it as a new submission. In my opinion, at least the main question, assumptions (or models), and results should be written very clearly.

The general motivation of this and previous work came from the principles of bottom-up cell design — how the quantitative properties of individual cell components and processes affect the growth and properties of cells.

The main purpose of the previous work was to improve the theoretical understanding of design principles concerning the self-reproduction system (SRS) (including theoretical doubling time limits and the role of ribosomes), whereas in this paper the objective is a more complete cell (SRS integrated with cell cycle) in the framework of the unit cell concept.

In short, in this study we tried to solve the following main problems associated with cell design principles:

- a. Expanding the self-replication analysis of cell components and proto-cells to complete cells by integrating the synthesis of cell components and CHD cell cycle theory. This would replace the doubling of only biomass with the self-reproduction and division of cells.*
- b. Expanding the analysis of empirical unit cell mass (M_u , central piece in CHD theory) to different unit cell parameters. This would enable to get deterministic M_u (and other unit cell parameters) and provide an opportunity to link it with actual molecular mechanisms behind M_u .*
- c. The analysis of cell load of unit cells. This would allow to understand the effect of the cell load on the quantitative properties (including maximal growth rate) of unit cells and give the theoretical basis for possible biotechnological applications.*

The take-home message in short form is the following:

- a. Simplified stoichiometric cell models were developed by integrating together models describing the self-reproduction of proto-cells and the unit cell concept from CHD cell cycle theory.*
- b. It was shown that cell load (cushioning protein) secures reasonable quantitative properties (cell size, slow growth) of unit cells.*
- c. The models allowed to determine and illustrate the main relations between the size and complexity of SRS, cell load and cell cycle*

parameters. Also, dependencies of the productivity of cushioning protein synthesis were explained.

We largely reworked the article text and included the problems and results described above in concise form to the Introduction (p 5, lines 107-119; p 6-7, lines 140-148). Also, we explained the general aim and background of both (current and previous) articles in Results and Discussion (p 7-8, lines 150-180). We created additional Supplementary Discussion 5.1 that describes simple relations between the key parameters: M_w , the number of ribosomes in the cell (N_{rs}), the number of cushioning proteins in the cell (N_{cp}), cell cycle length of unit cell (t_{cd}).

We largely rewrote and extended the whole section "Description of models" including the overview of model parameters, balance types in the models (equation types), differences between the models, etc. The purpose of that rewritten section is to give a general overview of all aspects of the developed models without too many technical details.

However, thorough detailed descriptions of all models are also available in Supplementary Information (in section Supplementary Discussion 5.10) including the descriptions of all equations, parameters and their values. We have used exclusively only simplified models in the current work but the general aim is to explain and develop a theoretical basis for larger models (strain-specific genome-scale models) which are useful in cell design and bioprocess optimization experiments. In our opinion, simpler models are useful for clearer presentation and explanation of the fundamental relationships.

We made a lot of additional changes in the resubmitted article to improve the readability and to also widen the discussion by including other references.

- 2. And I cannot believe that people can read papers with tons of abbreviations. It was a big surprise for me that more than 10 pages are dedicated to the description of abbreviations, symbols and definition of terms in SI text. Why do authors need so many symbols and abbreviations? For example, what is the point of abbreviating ribosomes as "rs"? It just confuses the readers, I would say.*

We now decreased the number of abbreviations significantly in the main text of the resubmitted article to improve readability. In addition, we decreased the amount of parameter symbols, terms characterizing parameter values and parameter units in the main text. Also, the formulations and descriptions of model parameters were shortened and made simpler in the main text.

However, in Supplementary Information we provide a complete technical description of all the used models, which is why the lists in Supplementary Discussions 1-4 include terms not mentioned in the main text. There are so many designations in those lists because the models contain so many designations (approximately 150 different parameters not to mention related abbreviations, indices, etc.). And it is not very reasonable to exclude those lists because the reader should have the opportunity to

understand the technical details of the models (necessary for reproducibility) if needed. We believe that the notations are foremost viable there. Perhaps, the concern about more than 10 pages of designations in Supplementary Information might actually not be that serious considering that regular descriptions of genome-scale models exceed our lists easily at least 10 times. It must be also noted that the reproducibility of the results is also important for the journal — for example the section Methods has unlimited space exactly for that reason (<https://www.nature.com/documents/CommsBio-file-checklist-revision.pdf>). The reason why these types of models are so large and have so many parameters is that they are mechanistic models. Unlike in statistical empirical models where the usual goal is to find the simplest model and to fit it to experimental data, mechanistic models usually describe the components of the system in detail to allow for calculating/simulating the (emergent) behaviours of the system. Thus, depending on the complexity of the system under study and the chosen level of abstraction for describing the components, the number of parameters and objects in such models can become huge, but for computer-based modelling this is usually fine. Concerning terminology, we agree that part of it is certainly original and it might contradict the terminology of other similar scientific fields. That is why we have included those long lists of terminology in Supplementary Information in addition to explanations in the main text — to minimize the possibility of misunderstandings.

The abbreviation for ribosome (rs) was inspired by the classical physiology studies of E. coli (for example Bremer et al., 2008) where ribosome was denoted by established r. Also, further derived rs is not uncommon in the literature (for example Baymukhametov et al., 2023; Kito et al., 2023). Still, we have now replaced rs in the main text of the resubmitted article with “ribosome” except in tables and figures.

3. Also, the authors are asking too much of the reader to refer to the SI text.

Indeed, the article was organized so that part of the content (relation between the models and published data about cell size measurements, dependencies of calculated parameter values, technical description of the models) and explanations (cushioning parameter, lists of designations) was available in the Supplementary Information considering the volume of this work. But it seems that we relied on this approach too much and we did not take into account possible difficulties for the readers. Based on that, we reorganized the whole text of the article and replaced referring sentences by longer and simpler explanations as much as possible. However, it is not possible to transfer everything from Supplementary Information to the main text due to its size. Therefore, the main text is a concise collection of main results, and Supplementary Information provides all explanations in detail, together with the technical descriptions necessary for the reproducibility of the work.

4. Overall, I see that some calculations have been done in the manuscript, but I cannot clearly see how these calculations have been done and what the values of the calculations are.

As already mentioned in the first point, we now rewrote the whole section “Description of models” and extended it to give a longer general overview of the used models. Calculations were carried out using Wolfram Mathematica and the corresponding code of the models is available in Supplementary File. Also, a general description of the calculation schemes (input and output parameters, variables) is now provided in the section “Description of models”. We created an additional Supplementary Figure 1 and Supplementary Discussion 5.1 that describe simple relations between key parameters as already mentioned in the first point. We shortly describe the calculation results based on standard input parameter values (visualized also in figures) but also based on published data from literature in the main text of the resubmitted article (p 19, lines 401-414; p 26-27, lines 522-553). Additionally, we stored the calculated values in more detail also in the tables of Supplementary Information (Supplementary Tables 4-9).

5. I am sorry that there are several points that I cannot follow up partly on due to my lack of knowledge, but I think that academic papers should be written with the authors' maximum effort to make the paper as readable as possible.

The authors feel also sorry that the submitted text was not sufficiently understandable. To improve this situation, we extensively reorganized and edited the main text of the resubmitted article. Certain sections were completely rewritten (“Cell cycle of unit cells”, “Description of models”) and others were supplemented with large segments of text (for example p 2, 5-7, 19, 26-27, 32, 36-38). Everything was edited to improve the readability.

Responses to reviewers' concerns

Reviewers' comments are written in Times New Roman and answers to reviewers are written in italicized Calibri.

Reviewer #1 (Remarks to the Author):

The resubmitted version of the article shows a notable improvement in most of the concerns.

The article is easier to follow, and each model is well justified. Although the article is relatively long, I think it explains most of the needed information. I just have a couple of recommendations:

1. Would you discuss typical parameters for the observed parameters in the main article as a reference to compare with the results. What is approximately value of the initiation mass in a bacterium (or cell volume assuming a typical cell density) and the typical number of ribosomes in a bacterium. The authors say 'slow' or 'small' but it is not clear you reference.

*The values of unit cell mass M_u (or initiation mass equal to $1 M_u$) estimated from experimentally determined data (cell size, cell cycle periods, growth rate) of exponential cultures of *E. coli* in different growth conditions (Bremer et al., 2008; Zhu et al., 2017; Si et al., 2017; Dai et al., 2018) are in the range of $1 \cdot 10^{-13} - 1 \cdot 10^{-12}$ g. Assuming usual cell density values, the corresponding cell volume range is $0.1 - 1$ fL. The numbers of ribosomes in the cell estimated from experimentally determined data (Bremer et al., 2008) depend on the growth rate and are in the range of $8 \cdot 10^3 - 7 \cdot 10^4$ ribosomes cell^{-1} (molar concentration range is $2 \cdot 10^{-4} - 4 \cdot 10^{-4}$ mol L^{-1}). The number of ribosomes in unit cells remains most likely below 10^4 ribosomes cell^{-1} . We have now added the corresponding values and references to the legend of Figure 1 (p. 11, lines 244-245), to the section "Calculated unit cells based on self-reproduction systems" (p. 15, lines 335-340), to Supplementary Tables 1-3 and to Supplementary Figs. 2-4, 11-14.*

Also, we now tried to make the various comparisons of cell parameter values throughout the text (including those that use words "slow" and "small") more clear (calculated values compared to literature data, calculated values of different models, calculated values based on different SRS complexities, calculated values for different growth boundary types, etc.). In the main text of the article, the comparison between models and literature is presented mainly in the sections "Calculated unit cells based on self-reproduction systems" (concerning Table 1) and "The effects of changing t_{CD} ".

2. I recommend using the same units of time and less significative digits. I would use the cell cycle time in hours if you were in the slow-growth regime. Experimentally does not make sense to have an accuracy of decimal of second.

Indeed, the use of hours as units for time seems to be reasonable for slower growth. However, there are several reasons for hesitation to completely replace seconds with hours. The main reason is that the models include the apparent working rates of enzymes and polymerases. The established and familiar unit of those parameters is molecules per second and not molecules per hour in the literature and also in BRENDA database (<https://www.brenda-enzymes.org/>). Although part of those parameters have generic values in models, it seemed also reasonable to maintain seconds as time units for readers who are familiar with apparent working rate units. More generally, the conceptual foundation of our models is constructing the cell from its molecular components, thus the emphasis is on thinking about the cell in the context of the properties of those molecular components, and on the molecular level of cellular processes seconds are usually a more natural unit than hours. Therefore, we retained seconds as time units but complemented them also with the corresponding values in hours in all texts, tables and figures where it seemed reasonable. Additionally, we decreased the number of significant digits in values of seconds by removing completely the fractional part after the decimal point. Also, we tried to decrease the overall number of significant digits during rounding of other parameter values as well.

3. I think it is needed to define a couple of concepts that are not clear specially to the community of mathematical modeling: Proto-cell, the role and mechanisms of ribosomes, cell load, cell cushioning.

Proto-cell in the context of our work can be defined as a cell or model that consists of only components of the self-reproduction system. These components are essential cell components that are necessary for the self-reproduction of the cell (DNA, RNA, membrane lipids, polymerases and enzymes, etc.). Proto-cell lacks all other cell components and processes which can be found in actual living cells and which perform various other functions not related directly to self-reproduction. We are aware that this term has been used quite freely in the literature to designate also first cells during evolution or simply primitive cells without certain important components but sometimes including also cell components that are not needed for self-reproduction (especially in bottom-up experiments of artificial cells). Respective short definition of our proto-cell has been included in Supplementary Discussion 4 (p. 13, lines 606-609) and in the Introduction of the main text we added a direct reference to that definition (p. 2, lines 31-36).

Ribosomes or more precisely ribosomal proteins are unique cell components because they are the only cell components that are able to self-reproduce themselves. Their self-reproduction can be described in a very simplified approach by the following formula:

$$t_{d_rs} = \frac{N_{rs} \cdot n_{rpc}}{N_{rs} \cdot k_{rs}} = \frac{n_{rpc}}{k_{rs}}$$

It appears that the doubling time (t_{d_rs}) of a ribosome is equal to the ratio of the number of amino acid molecules in the ribosomal protein complex (n_{rpc}) to the apparent working rate of the ribosome (k_{rs}). Using the standard input parameter values of SSUCMs, the value of t_{d_rs} calculated from the formula above is

approximately 6 minutes. It also appears that the doubling time does not depend on the number of ribosomal proteins (because N_{rs} cancels out in the formula above). If sets of independent ribosomal proteins could be considered simple proto-cells there is no limitation on the size of such proto-cells. However, the synthesis of other proteins by ribosomes means that there will be additional products in the numerator of the formula increasing the value of the respective doubling time. Therefore, the inclusion of other cell components always prolongs the doubling time because they are not synthesized by themselves. According to the simplest description, the fraction of ribosomal proteins in the total protein content decreases linearly with growth rate decrease and does not depend on other parameters beside n_{rpg} , k_{rs} and t_{CD} . The role and mechanisms of ribosomes are shortly described in section "Cell cycle of unit cells" (p. 13, lines 282-293), the dependence between growth rate and the fraction of ribosomal proteins in the total protein content is visualized in Supplementary Figure 9 and Supplementary Eq. (6). A thorough presentation on this subject can be found in (Abner et al., 2023).

Cell load in the context of our work can be defined as a collection of cell components that are not directly necessary for cell self-reproduction (i.e. not belonging to the SRS of the cell) and of unused (i.e. currently inactive) cell components from the SRS of the cell. In the current work, cell load is manifested as generic cushioning protein in the used SSUCMs. Possible examples of cell load in real prokaryotic cells might involve different proteins with various secondary functions in terms of self-reproduction, reserve materials, just-in-case pathways and transporters, idle ribosomes, recombinant proteins, etc.

Respective short definition of cell load has been included in Supplementary Discussion 4 (p. 13, lines 583-588). The essence of the cell load has been also described in Introduction with a reference to Supplementary Discussion 4 (p. 4-5, lines 91-109) and more thoroughly in section "Introducing the cushioning protein" (p. 17-18, lines 372-403) and in Supplementary Discussion 5.5 (p. 32-33, lines 184-229).

Cell cushioning is a central concept in the SCM framework and is interrelated with the cell load concept. It is assumed that there is a growth-dependent physiological parameter(s) in the cell that secures reasonable cell sizes, cellular concentrations and slower growth rates which is not always possible with only SRS doubling. Indirectly it represents also the united functioning of cell load components. Cushioning can be viewed as a common denominator to various categories of cell components including special secondary cellular functions (regulatory, structural, homeostasis, etc.), reserve materials and unused SRS parts (idle ribosomes, metabolic pathways and transporters) which allow to respond flexibly to various changing growth conditions. In SSUCMs, the cushioning has been achieved by the generic protein in the model cell that is not directly necessary for the self-reproduction of the cell and does not have any specific function in the cell.

Respective short definitions of cushioning parameter and cushioning protein have been included in Supplementary Discussion 4 (p. 13, lines 595-605). The essence of the cushioning has been also described in section "Introducing the cushioning protein", in Discussion (p. 35-37, lines 744-797) and more thoroughly in Supplementary Discussion 5.4 (p. 30-32).

- As an optional change, I think it is possible to approach the analysis instead using number of molecules, molecule concentration. This to have a better understanding using traditional units.

There are several reasons why we have clung to such unusual units as numbers of molecules instead of proper concentration units. One of the main (although subjective) reasons is philosophical. In our opinion, molecule numbers per cell provide the most adequate description for cell-centered approach in modelling whereas concentrations rather refer to biomass. When talking about single cells, it is useful to talk separately about how many of various molecules there are inside a single cell, and about what is the volume of that single cell, because it is insightful to see how these values can change separately. We belong to the cell-centered modelling community. Also, the described dependencies between parameters are completely different in the case of per cell units compared to per mass/volume units. The essentially linear relationships (for example Figure 2, also below) between molecule numbers and cell mass are transformed to completely non-linear relationships (for example Supplementary Fig. 5, also below), calculations based on constant molecule number values correspond to variable concentration values (for example Supplementary Fig. 1ab), etc. Therefore, we are afraid that such replacement of units would require a serious rethinking and simple dependencies would become more complicated and less insightful.

Figure 2.

Supplementary Fig. 5.

On the other hand, we acknowledge that concentration units are more familiar for a wide range of readers. Also, alternative units are sometimes useful for representation (for example a completely linear relationship between ribosomal protein fraction and growth rate instead of doubling time). Therefore, we reached a compromise and retained molecule number units but complemented them with molar concentration units in all texts and tables where it seemed reasonable. As it was not reasonable to put both units into the same figures, parallel figures were created for molar concentration units (Supplementary Figures 1, 5, 15, 21, 23, 25, 27, 38, 40, 42, 44, 46, 48, 57).

5. I would finally recommend to add a conclusions paragraph at the end of the discussion to highlight quickly the main findings.

We have now added a new paragraph consisting of the main conclusions at the end of the Discussion section (p. 41, lines 884-895) to emphasize better the main results of the paper.

Reviewer #2 (Remarks to the Author):

The authors have made improvements in the manuscript's writing compared to the previous version.

1. However, the text remains quite complex and does not adhere consistently to the principles of academic writing, such as dedicating each paragraph to a single topic. As a result, the manuscript is still difficult to read.

We have now further edited the text and tried to improve the readability. Among other changes we reworked the contents of paragraphs (changes too numerous to report in detail). We hope that the principle of a single topic per paragraph is now better adhered to in the paper.

2. Nonetheless, the revisions and responses have clarified the manuscript's focus for me. My major concerns are as follows:

1. The model description should be clearly presented within the main manuscript rather than relegated to the Supplementary Information (SI).

We have organized the texts involving the descriptions of models so that the general overview is presented in the main manuscript. It seemed reasonable to collect everything into a section "Description of models" rather than distribute them between various sections. The section "Description of models" should provide a concise but still comprehensive description of the used models, their cell components and processes, base assumptions, equation types, input and output parameters, calculation schemes and other relevant issues. In addition, the used models are briefly mentioned in the section "Results and Discussion" (p. 8-9, lines 178-197). The unit cell concept and the cell cycle model are described in the section "Cell cycle of unit cells" (p. 9-13, lines 204-281). The purpose of those sections is to give a general overview of all aspects of the developed models without too many technical details. However, thorough detailed descriptions of all models are also available in Supplementary Discussion 5.10, including the descriptions or references of all equations, parameters and their values. The purpose of that part is to give the opportunity to understand the technical details of the models (necessary for reproducibility) if needed.

We pondered on the possibility of merging both descriptions into the main manuscript but we realized that the volume of the manuscript text would have been extended enormously. However, we tried to ensure that the general overview in the main manuscript answers all principal questions about differences of models, main cell components and processes, about which models were used for which calculations, etc. If it still fails to do so, then we can extend it even further, of course.

3. For instance, in line 297, the authors state: "First, we calculated the μ and other parameters of UC at approximate $t_{CD} = 3519.84 \pm 296$ using SSUCM-SRS-M and SSUCM-SRS-R, which are just SRSs at unit cell condition." This explanation is too brief and lacks sufficient detail.

The respective introductory sentence at the beginning of the section “Calculated unit cells based on self-reproduction systems” refers to calculations made at relatively slow growth rate ($t_{CD} = 3520$ corresponds to the unit cell condition based on the standard input parameter values of our models) using models that include only SRS components.

To improve the understanding of this and other similar cases, we now further expanded the paragraphs explaining model parameters and calculation schemes in section “Description of models” (p. 48-50, lines 1052-1094). Most of the input parameters (for example the apparent working rates of polymerases and enzymes, the number of reactions in pathways, the energy costs of polymerisation and biosynthesis, the numbers of monomers in cell components, the surface areas of membrane components, the masses of cell components, etc.) have constant values in calculations. The exceptions are t_{CD} , cell division time t_D , the apparent working rate of replisome k_{dp} and the number of cushioning proteins in the cell N_{cp} (which in some of our calculations is actually an output parameter). They are used in some calculations as independent variables or some of them have experimentally determined values. Also, in one exceptional case the apparent working rate of the ribosome k_{rs} was explicitly lowered to demonstrate how this would allow the model to better match certain experimental results (but it is important to note that this was an exception: in most of the calculations the input parameters are kept strictly constant at their standard biologically feasible values, which ensures that we cannot use the seemingly large number of parameters to just mathematically fit the model to any possible data). All other parameters (the numbers and concentrations of cell components, fluxes, compositions, geometric dimensions) are output parameters and their values were calculated with the models. A list of which calculations were performed in which sections of the manuscript is now also provided in the section “Description of models”:

- a. Calculations based on constant standard values of input parameters in section “Calculated unit cells based on self-reproduction systems” using models SSUCM-SRS-M and SSUCM-SRS-R.*
- b. Calculations based on the variable N_{cp} and constant standard values of other input parameters in section “The effects of cushioning protein” and Supplementary Discussion 5.7 using models SSUCM-M and SSUCM-R.*
- c. Calculations based on the variables N_{cp} and t_{CD} and constant standard values of other input parameters in sections “The effects of changing t_{CD} ” and “Productivity of cushioning protein synthesis in unit cells” using models SSUCM-M and SSUCM-R.*
- d. Calculations based on constant standard values of input parameters and experimentally determined values of t_{CD} in Supplementary Discussions 5.2, 5.6 and 5.8 using models SSUCM-M and SSUCM-R. In addition, there are a few special cases such as calculations based on changed values of k_{rs} and data of (Si et al., 2017) (mentioned in section “The effects of changing t_{CD} ” and in Supplementary Discussion 5.8).*

4. 2. The stated objectives of the manuscript—to "expand the analysis," "provide an opportunity to link..." and "give the theoretical basis" (lines 109-119)—are vague and do not constitute clear scientific questions. It is always possible to argue the expansion of any analysis or the provision of any opportunity, for instance just by adding one more data point. The authors clarify what question is revealed by this expansion.

Although the main objective of the manuscript was the development of a modelling framework for growing cells, there are indeed several theoretical biology questions that can be brought up. The following list contains some of the specific questions we were interested in:

- a. *Given that the previous analysis (Abner et al., 2023) showed that the doubling of SRS is in most cases extremely fast with a narrow range of doubling time values except for very small-sized SRSs, how are the actual biological larger cells able to grow with a very wide growth range? The integration of SRS doubling with the CHD cell cycle theory makes it possible to answer this question.*
- b. *Where are the growth boundaries of unit cells? CHD theory provides strict dependencies between growth rate and cell size based on the values of M_u and cell cycle periods, but CHD theory does not provide any rules for the determination of (minimal/maximal) M_u values. Our models allow to treat M_u not as just an empirical parameter but as a deterministic parameter that is linked with actual molecular mechanisms and with the properties of cell components, and thus our models allow to find also the theoretical growth boundaries of cells (such as the minimal/maximal values for M_u and for the cell doubling time).*
- c. *Where is the optimal region of cell load synthesis productivity in unit cells? While it is already common knowledge that the synthesis of other proteins (including recombinant proteins) decreases the growth rate, the relation between synthesis productivity and cell size or growth rate has not been established. Our analysis of the cell load of unit cells provides a way to determine the optimal region of cell load synthesis productivity.*

The stated objectives in the section "Introduction" are now amended with the specific questions (p. 5-6, lines 110-135).

5. 3. The authors claim that constructing a minimal model is not their goal. However, the model presented is neither detailed (like a whole-cell model) nor minimal. If a model is detailed and realistic, its parameters should be empirically derived. Conversely, if a model is minimal and coarse-grained, parameter selection can be somewhat arbitrary, though the model's flexibility is limited by the small number of variables. Both approaches offer valuable insights. However, an intermediate model, which sits between realistic and minimal, risks reproducing any result due to the lack of constraints on the number of variables and parameter choices. I find it challenging to see the value in the outcomes from such a model.

The claim about the minimal model not being our goal was associated with the explanation of design parameters and why we need 4 models instead of 1 (to illustrate different complexities of SRSs and the availability of cushioning loads). The primary purpose of the current work is to improve the theoretical understanding of the unit cell concept in the context of cell design by integrating together SRS and cell cycle. The developed models are mechanistic models. Unlike in statistical empirical models where the usual goal is to find the simplest model and to fit it to some particular experimental data, mechanistic models usually describe the components of the system in detail to allow for calculating/simulating the (emergent) behaviors of the system. Most importantly, unlike in statistical empirical models where the model parameters can be freely adjusted to make the model fit the data, our mechanistic models have almost all of the input parameters strictly fixed to constant values that are not chosen arbitrarily by us but are based on actual biological values from experimental data in the literature. The only exceptions are t_{CD} (cell division time t_D , the apparent working rate of replisome k_{dp}) and the number of cushioning proteins in the cell N_{cp} , which were used in some calculations as independent variables, yet again not for fitting but for illustrating how the change of their values changes the cell. There was only one exceptional case where we explicitly demonstrated how picking a different value for the apparent working rate of the ribosome k_{rs} would allow us to get a better fit with experimental data compared to using our standard fixed parameter values.

Principally, we agree that certain types of intermediate models have elevated ability to reproduce data (in comparison to the simplest models) and also to falsely reproduce data. But based on our experience, the latter phenomenon in models is largely caused by having a large number of variable parameters and also by the functioning of a complicated metabolic network (even if it is not as complete as in genome-scale models) which should be seriously underdetermined (degrees of freedom) to cause various artificial flux cycles. However, we consider all the 4 models that we use in this paper to be very strongly simplified coarse-grained models (in the class of the simplest possible models that allow illustrating how the basic fundamental building blocks and processes of the cell work together) and not intermediate models because:

- 1. They contain only a very small set of fundamentally important simplified cell components and processes (Abner et al., 2023).*
- 2. Their metabolic networks are strictly determined (i.e. the number of metabolites is equal to the number of reactions) which means that the mathematical structures of those models are very rigid.*
- 3. Almost all of the input parameters are constants, as explained above, which again significantly constrains the mathematical solution space.*

Basically, this can be verified by the calculation results of SSUCM-SRS-M and SSUCM-SRS-R (summarized in Table 1, Supplementary Tables 13-14) which were not able to reproduce known experimentally observed data (cell size, numbers of ribosomes, etc.) at all (the calculated values were not even close). Also, SSUCM-M and SSUCM-R at standard input parameter values were not able to reproduce measured RNA/protein values (Supplementary Figure 9, below). In our opinion, such simplified models are usually not suitable for exact reproduction of experimental values but they can be useful for the reproduction of general trends and for the explanation of fundamental

relationships in the cell. For example, the used SSUCMs can reproduce the linear relation between growth rate and ribosomal protein fraction (designated with a red line in Supplementary Figure 9).

Supplementary Fig. 9.

Overall, the idea was to construct as simple models as possible because simplified models produce simpler results and they are more understandable for readers. As already mentioned earlier, the main goal was the improvement of theoretical understanding and not the exact reproduction of available data. We understand that our models are larger than some of the simplest models in the literature and, therefore, they might be potentially more “capable”. We needed models of this exact size because we wanted to describe also certain cell components and processes that are usually not included in the simplest translation or cell cycle models. Otherwise, it would have not been possible to show relations involving cell geometry, DNA, membrane lipids, etc. But we still think that all our simplifications and strong constraints together ensure that the solution space of our models is heavily constrained and does not allow for fitting to whatever data.

6. 4. This point relates to my 3rd comment. The introduction of cushioning parameters appears to be a way to align the calculation results with existing literature. While I acknowledge that abstract proteins, like cushioning proteins, are sometimes introduced in models, particularly in proteome-allocation physiology research, these models are usually minimal. For example, in Scott et al., Science (2010), the introduction of class-Q proteins is not crucial to the

main argument.

For a start, we consider the developed SSUCMs to be in the same category of the simplest models as the minimal models from literature as discussed in the previous point.

Secondly, the introduction of cushioning is indeed a possible solution how to force calculation results to coincide with existing literature. However, we were not introducing some arbitrary free parameter to allow for artificial mathematical fitting. The starting point of the calculations was based on models that did not include cushioning (SSUCM-SRS-M, SSUCM-SRS-R). Alas, and as mentioned before in the previous point, those models were incapable of describing the growth of actual living cells because the calculated cells were either too small or were growing too fast (summarized in Table 1, Supplementary Tables 13-14).

*Certainly, it is theoretically possible to get an alignment between literature and the calculations with those models if the value(s) of some input parameter(s) is/are varied during calculations to obtain a reasonable fit. Part of the physical/molecular parameters (for example amino acid or nucleotide sequence lengths, masses of cell components, etc.) are rather constant for a specific strain or their values are varying only insignificantly inside single species (for example the conserved structure of ribosomes). The most reasonable candidates for varying would be the apparent working rates of catalysing cell components and the size of the metabolic network (the number of reactions). The increase of reaction numbers will increase the numbers of necessary enzymes in the cell and eventually will increase the overall cell size. In the case of our minimal media example, this will happen if the number of reactions would be increased approximately from 1000 to 1800. Whereas this is somewhat conceivable based on the sheer number of reactions in genome-scale models of *E. coli* (<https://www.ebi.ac.uk/biomodels/MODEL1108160000>), it must be taken into account that a considerable number of those reactions are located in the secondary metabolic pathways with small fluxes (and thus with small numbers of enzymes). The important core metabolism is much smaller (Hädicke et al., 2017). The increase of reaction numbers is even more dubious in the case of growth on rich media because on rich media monomers are consumed from the environment instead of using long biosynthesis pathways. Of course, this does not mean that the corresponding biosynthesis enzymes are not synthesized anymore. But if they are synthesized and no biosynthesis is needed, then such pathways and enzymes are not directly necessary for self-reproduction. Therefore, according to the definition they should be considered as cell load and cushioning parameters.*

Alternatively, it is possible to increase the size of cells also by decreasing the apparent working rate of enzymes (generic values in models) assuming that the values of these parameters are very much growth-dependent (lower values during slower growth). In the case of minimal media, the apparent working rate of enzymes should fall 50 %, but in the case of rich media the decrease can be associated only with the apparent working rates of polymerases (membrane proteins would be worse candidates because their lower rate would lead to the membrane limitation appearing even earlier). For example, the decrease in the apparent working rate of the ribosome should be approximately 10 times which is not a common figure. If it is possible, then it actually refers to a considerable number of unused ribosomes (i.e., a

small number of normally active ribosomes together with a larger number of idle ribosomes would result in a low apparent working rate over the whole population of ribosomes). Again, these unused ribosomes must be considered as cell load according to the definition.

Mathematically it is possible to change a combination of several parameters to avoid the most extreme situations but perhaps the main reason to abandon the SSUCMs without cushioning was that there are still many other proteins in actual living cells that are not involved in self-reproduction but are instead fulfilling structural, housekeeping, regulatory and other functions. Based on that, the most reasonable solution seemed to be to include the synthesis of cushioning protein which represents the united set of unused SRS components and proteins not needed for self-reproduction. It is also a usual case in metabolic models that some part of the measured proteome must be dropped from the model (for example annotated as hypothetical proteins) or the leftover must be somehow taken into account globally. Abstract proteins in the form of “leftover” proteins (not described specifically in models as they do not belong to the metabolic or polymerisation network) can be also found in large genome-scale models (for example in (Elsemman et al., 2022)). Basically, the global protein balance of practically all genome-scale models that take into account experimentally determined proteome data must also include proteins that remain outside of the defined metabolic network reactions. In addition, we can not forget the serious experimental limitations of membrane proteomics. Certainly, it is possible that the cushioning mass is formed by something else than accumulating proteins but currently we wanted to cover only a single example and not all possible cases.

The goals of our work (estimation of unit cell parameters) were different compared to the work of (Scott et al., 2010) where the main goal was to elucidate proteome allocation and ribosomal fraction growth laws. Indeed, Q- and also U-class proteins from their work are not really necessary to explain the essential changes in ribosomal protein fraction and RNA/protein ratio because only a single non-ribosomal protein fraction would be enough.

The overlapping part of QPR and SSUCM models is R-class proteins. SSUCMs do not have a fixed Q-sector (because the contents of all proteins, including also the cushioning protein, depend on t_{cd}) and instead of a single P-class there are different variable proteins. The sum of their contents is naturally inversely proportional to the ribosomal protein fraction content but individually these relations vary. Whereas cushioning protein would not be needed to explain the ribosomal protein fraction which is independent of molecule numbers (Supplementary Eq. (6)), it was needed to explain the discrepancies of the values of other parameters including M_u .

7. 5. The results presented seem mathematically trivial. The authors should emphasize the biological novelty of their findings, perhaps by demonstrating that the model fits well with a substantial portion of the data. However, this is not adequately presented in the manuscript (I checked SI. But I cannot see the figures supporting the model “prediction”. The figure just shows the calculated values by the model, rather than comparing the model prediction and data?).

The comparison of model calculations of unit cell parameters (based on SSUCM-M and SSUCM-R) and literature data had been already included to the previous resubmitted version of the manuscript but it seems that the presentation was not adequate and easy to find. Therefore, we have now improved the presentation:

- a. It was possible to estimate the dependencies of experimentally determined (various growth regions, based on data of (Volkmer et al., 2011), (Zhu et al., 2017), (Si et al., 2017)) and calculated (unit cell condition) data points of cell size (Supplementary Figs. 11-13, also below). The comparison showed that calculated data points reproduced sufficiently the trend of experimentally determined data points at different growth rates.

Supplementary Fig. 11

Supplementary Fig. 12

Supplementary Fig. 13

b. Similarly, it was possible to estimate the dependencies of experimentally determined (various growth regions, based on data of Bremer et al., 2008) and calculated (unit cell condition) data points of numbers of ribosomes (Supplementary Fig. 14, also below). The comparison showed that calculated data points reproduced sufficiently the trend of experimentally determined data points at different growth rates.

Supplementary Fig. 14

- c. Previously, averages of data points from (Si et al., 2017) were used in the resubmitted version of the manuscript but now we have used all the experimental data points in order to have more data points in unit cell region. All the corresponding texts, tables and figures have been updated. The comparison of calculated and measured values of the ratio of M_{rna}/M_{prot} showed that the general trend of the dependence between M_{rna}/M_{prot} and growth rate was similar (Supplementary Figure 9, also below). However, the numerical values of the calculated M_{rna}/M_{prot} in the case of standard input parameter values were considerably lower than the experimentally determined values. The reasons for such a discrepancy might form a long list — the inability of simplified models to precisely describe the specific strains (metabolic networks, missing cell wall components), the assumption of cushioning mass being composed of only cushioning protein and excluding other possibilities like inactive ribosomes, growth-dependent cell parameter values, etc. In our opinion, as already mentioned earlier, such simplified models are usually not suitable for exact reproduction of experimental values but they can be useful for the reproduction of general trends and for the explanation of fundamental relationships in the cell.

Supplementary Fig. 9

The comparison of t_d and t_{CD} values showed that there are a few experimental growth points that are sufficiently close to unit cell condition and thus good candidates for further analysis: MG1655 strain growing on semi-complex media and both *E. coli* strains growing on minimal media. We demonstrated that the calculated value of the ratio of M_{rna} to M_{prot} could be made to approximately match the measured value by, for example, lowering the value of k_{rs} from 20 to the range of 3 – 7 molecules (aa) $s^{-1} rs^{-1}$, (Supplementary Fig. 10, also below). The fitted values of k_{rs} are presented in Supplementary Table 9.

It must be stressed that the standard value of k_{rs} in SSUCMs is probably the possible upper limit which is achieved during very fast growth. Slower growth, however, is characterized by lower k_{rs} values according to the literature. Most probably, the real k_{rs} values corresponding to the data of (Si et al., 2017) are somewhat higher than found in our simple illustrative fitting because other abovementioned factors influence the calculation results as well. In order to clarify the effect of different factors, strain-specific models are needed.

It must be stressed that this is the only example of changing the value of input parameters beside independent variables (t_{CD} , N_{cp}) in the current work. All other calculations were carried out using the standard parameter value of k_{rs} . The previously shown calculated

ribosome numbers in Supplementary Fig. 14 fitted more or less to the same curve without any changes in k_{rs} value (using our standard value $k_{rs} = 20$). Therefore, there seem to be considerable differences between different datasets in the literature which makes it impossible to always use only a single set of input parameter values for the model if the goal is to fit experimental data.

Supplementary Fig. 10

The calculated values of the same UC parameters based on the standard value of k_{rs} are presented in Supplementary Table 10, and the values based on the previously fitted k_{rs} values are presented in Supplementary Table 11.

8. 6. Overall, the authors explore the model's dependency on the parameters, but I cannot see why we need to know it, because the reliability of the model to understand biology is not presented.

Concerning the reliability of the model:

- a. As can be seen in the previous point above, the developed SSUCMs are able to reproduce different trends of experimentally determined parameter values including cell size, numbers of ribosomes and M_{rna}/M_{prot} . It must be noted that the models in the current work describe only unit cells ($t_d = t_{CD}$, only a single genome at t_0), whereas the data from the literature were collected from cells growing in batch or turbidostat cultures which means that usually $t_d \neq t_{CD}$ and overlapping genome replications happened. Therefore, it was not possible to carry out full reproduction of data. In addition, the

- simplified models are not suitable tools for exact reproduction of experimental values which preferably requires strain-specific models.*
- b. Although the calculated ranges were intentionally overextended (in terms of t_{CD} and N_{cp} values) to illustrate maximally wide theoretical growth boundaries, the obtained calculation results cover also the physiologically relevant region. The calculated values of cell parameters including M_u , the number of ribosomes, DNA and RNA contents within the physiological region ($M_u < 10^{-11}$ g, Supplementary Tables 5-8) correspond to the reported experimental values.*
 - c. Part of the balance equations in the developed SSUCMs are similar to established equation types already published in the literature (ribosomal protein and RNA polymerase translation balances, RNA transcription balance (Abner et al., 2023), CHD cell cycle, etc.).*
 - d. The values of the used input parameters are strain-specific (label specific, precise), species-specific (labels approximate and average) or at least inside the reasonable biological range (label generic). The values were generally constant and not variable except independent variables (t_{CD} , N_{cp} , but these were used not for fitting but for illustrating how the change of their values changes the cell).*

We have explored the dependencies of various (in our opinion important) parameters because this lays the foundation for the understanding of how cells are designed. It is possible that some of the illustrated relations might be different in some specific living cells due to the simplifications of the models, but then the comparison of the models to those particular cells would allow to find the reasons for such differences. However, as far as we can see, most of the fundamental relations that our models illustrate are in a reasonably good agreement with experimental data.

9. Regarding the authors' rebuttal letter: They respond by stating that "rs" is a common abbreviation in the field and that "Genome-scale models are more complicated than ours." I encourage the authors to be more constructive in their responses. My point is that readers are required to remember numerous abbreviations. Considering the cognitive load on readers, what justifies the use of the abbreviation "rs"? My intention in discussing the number of abbreviations was to highlight that the manuscript is difficult to read. Readability and reproducibility are not mutually exclusive. Many well-written genome-scale studies effectively present their questions, methods, results, and conclusions without relying on an excessive number of abbreviations. Detailed information can be provided separately for those interested in reproducing the results. Arguing that "this approach is acceptable because another paper has done it" is not a constructive rebuttal.

It seems that there is some kind of misunderstanding, perhaps due to our non-native language or cultural differences. We tried to explain in our response letter that we considerably reduced the amount of designations (including abbreviations) in the main text of the manuscript, that the long lists of designations in the Supplementary Information are mainly for the technical model descriptions (which are also located in the Supplementary Information) and what was the origin of the rs abbreviation. Also, the lists of designations in the Supplementary Information should not disturb reading

the main text and the rs abbreviation was largely removed from the main text. We certainly did not ask to keep as many abbreviations as possible in the main text. If it seems that there are still too many designations in the main text, then we can replace more of them. We did not replace all designations because we are quite sure that replacing all short designations with their full text equivalents would also cause serious readability problems, so we did our best to try and find a suitable balance. We fully agree that this point of balance might be different for different readers.

Responses to reviewers' concerns

Reviewers' comments are written in Times New Roman and answers to reviewers are written in italicized Calibri.

Reviewer #1 (Remarks to the Author):

Most of my concerns were addressed.

1. The remaining concern is related to the time units. the authors discussed that the relevant time scale was seconds. However, Figure 6 and 7 have units of hours.

Indeed, the unit of the specific productivity of cushioning protein synthesis (Q_{cp} , molecules (cp) (g (cell))⁻¹ h⁻¹) visualized in Figures 6-7 does not fit conceptually to the developed modelling framework of SSUCMs. The main distinctive feature of SSUCMs is the description of self-reproduction of single cells and their cell cycle. According to this feature, the most appropriate unit for the description of the amounts of cellular components is "per cell". The most suitable unit for self-reproduction time is second because the cell cycle of a single unit cell is characterized by cell cycle length and second is an established unit for apparent working rates of enzymes and polymerases.

However, in the article we are analysing Q_{cp} with the specific goal to demonstrate the practical relevance of our modelling framework to the biotechnology industry, and, for this reason, we decided to make an exception for Q_{cp} and use the established units of the field of biotechnology. Productivity in biotechnology is usually either volumetric or per mass, i.e., the amount of product formed per volume or mass per time. Because the growth of cells is described in biotechnology almost exclusively as the doubling of biomass and not individual cells, the productivity parameter is directly proportional to specific growth rate of cell culture, and the established unit for specific growth rate is h⁻¹.

We have now included an additional explanatory sentence to the section "The productivity of cushioning protein synthesis in unit cells has optima" (p. 24, lines 562-566).

2. I would recommend some other minor style changes.

Use subsections with title being a summary of the main finding using a phrase. This helps to follow better the article and motivate the reader to follow the reading.

We have now changed the titles of most of the subsections of the Results section so that the titles describe the summary of the main results in the form of a phrase:

- a. Calculated unit cells based only on self-reproduction systems are problematic (p. 13, lines 293-294).*
- b. Cushioning protein ensures physiologically reasonable cells (p. 16, lines 378-379).*

- c. *Changing t_{CD} has major effects on properties of unit cells* (p. 18, line 425).
- d. *The productivity of cushioning protein synthesis in unit cells has optima* (p. 23, lines 554-555).

Also, we now divided the section “Changing t_{CD} has major effects on properties of unit cells” into two parts and titled the second part as “Calculations reproduce reasonably the experimentally determined trends in unit cells” (p. 21, lines 505-506).

- 3. Given the length of the article, it is good to use a concluding sentence at the end of each section and a motivation to the next one such as the reader have clear why you structured the article the way you presented and what are the take-home messages. It will not be possible to read the article quickly, therefore you need to provide the reader pauses along the text.

We have now included additional sentences at the ends of the following subsections of the Results section to emphasize better the main results of each subsection and to provide transition to the next subsection:

- a. *“Cell cycle of unit cells”*: p. 12, lines 290-292.
- b. *“Calculated unit cells based only on self-reproduction systems are problematic”*: p. 14, lines 338-340.
- c. *“Introducing the cushioning protein”*: p. 16, lines 374-377.
- d. *“Cushioning protein ensures physiologically reasonable cells”*: p. 18, lines 423-424.
- e. *“Changing t_{CD} has major effects on properties of unit cells”*: p. 21, lines 501-504.
- f. *“Calculations reproduce reasonably the experimentally determined trends in unit cells”*: p. 23, lines 551-553.
- g. *“The productivity of cushioning protein synthesis in unit cells has optima”*: p. 25-26, lines 606-609.

Reviewer #2 (Remarks to the Author):

The authors have provided sufficient responses and revisions to clarify the issues and improve the overall quality of the manuscript. I find the current version satisfactory and recommend it for acceptance.